# Mantle exhumation at magma-poor rifted margins controlled by frictional shear zones

Thomas Theunissen [1✉] & Ritske S. Huismans [1]

The transition zone from continental crust to the mature mid-ocean ridge spreading center of the Iberia-Newfoundland magma-poor rifted margins is mostly composed of exhumed mantle characterized by highs and domes with varying elevation, spacing and shape. The mechanism controlling strain localization and fault migration explaining the geometry of these peridotite ridges is poorly understood. Here we show using forward geodynamic models that multiple out-of-sequence detachments with recurring dip reversal form during magma-poor rifting and mantle exhumation as a consequence of the strength competition between weak frictional-plastic shear zones and the thermally weakened necking domain beneath the exhuming footwall explaining geometry of these peridotite ridges. Model behaviour also shows that fault types and detachment styles vary with spreading rate and fault strength and confirm that these results can be compared to other magma poor passive margins such as along Antarctica-Australia and to ultra-slow mid-ocean spreading systems as the South-West Indian Ridge.

[1] Department of Earth Science, University of Bergen, Allegaten 41, Postboks 7803, N-5020 Bergen, Norway. ✉email: thomas.theunissen@uib.no

The Iberia-Newfoundland magma-poor passive margins exhibit a wide transitional domain (up to 170 km) mostly composed of exhumed mantle exposed on the seafloor and episodic magmatic activity expressed in thin to normal oceanic crust between the distal rifted continental crust and the oceanic domain[1,2]. The transition from magma-poor spreading and mantle exhumation to the establishment of a stable mature mid-oceanic spreading center, defined by standard oceanic crustal thickness in the range 4–8 km and locally exhumed mantle domes for slow and ultra-slow spreading rates[3–5], appears to be gradual with progressive increase of magmatic addition towards the distal domain[6,7]. The transitional is to first order symmetric on the conjugate margins[8]. The distal continental margin adjacent to the transitional domain involves crustal detachment faulting resulting from the coupling of crustal thinning and mantle exhumation during the last stage of continental rifting[9]. The transitional domain can be compared with melt-poor ultra-slow mid-ocean spreading ridges where part of the extension is accommodated by high-offset detachment faults with footwall exhumation of mantle rocks[10]. At ultra-slow mid-ocean spreading ridges, this exhumation involves successive detachment faults that alternate polarity and that have a lifetime of 0.7–1.5 Myr[11–13]. This mode of faulting has first been described as flip-flop detachments for the conjugate Iberia-Newfoundland passive margins[14]. Despite the recognition that efficient weakening mechanisms, a thin brittle layer, and low magma supply favor long-offset detachment faults, with dome-shaped exhumed fault footwall and the characteristic shape of the fault breakaway[15–17], the processes explaining strain migration and the position of each new fault are poorly understood[4,5,8,12,18–21]. Footwall strength reduction due to magma supply, hydrothermal activity, or serpentinization, as well as migration of the exhumation point away from the thermal necking axis are commonly proposed explanations for strain migration into the footwall of the active detachment fault. Previous modeling studies on large offset detachment faults focused on the relative importance of magmatic versus tectonic spreading[5,17,22], melting[16,23], and complex shear zone weakening mechanisms including serpentinization and grain size reduction[12]. However, none of these studies fully reproduced the observed characteristics comprising offset, topography, and wavelength of peridotite ridges, and inferred thickness of the brittle layer in these systems. The mechanism controlling the mode of faulting, strain migration during mantle exhumation, and formation of peridotite ridges such as along the Flemish Cap-Galicia conjugate margins with highs and domes observed in this domain is consequently poorly understood.

We focus on the northernmost part of the Iberia-Newfoundland conjugate rifted margin system (Fig. 1). High-resolution conjugate seismic reflection and refraction profiles on the Flemish Cap (e.g., SCREECH-1)[6,24] and Galicia conjugate

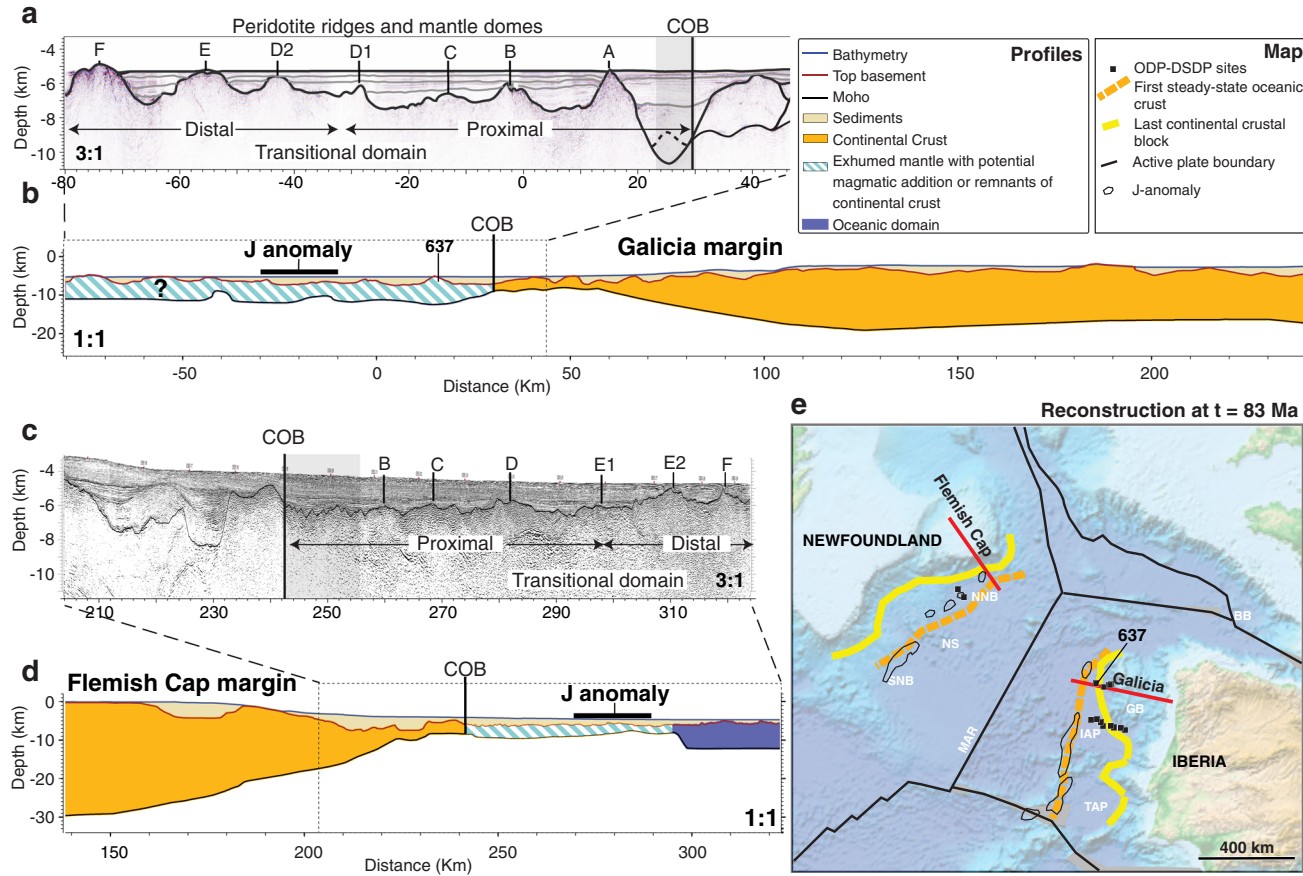

**Fig. 1 Iberia/Newfoundland conjugate margins. a** Distal part of the seismic reflection profile WE-1 on Galicia side[26]. **b** Large-scale geophysical cross-section of the Galicia margin based on composite ISE-1/WE-1 refraction seismic profiles[7,25,26]. Proximal domain not shown. **c** Distal part of the seismic reflection profile SCREECH-1 on Flemish Cap side[6]. **d** Large-scale geophysical cross-section of the Flemish Cap margin based on the SCREECH-1 deep seismic refraction profile[6,24]. **e** Map at time C34n (83 Ma) with present-day ETOPO-1 relief, based on G-Plates reconstruction of Matthews et al.[64]. Magnetic J anomaly from EMAG2v3 data set[48]. Note that the magnetic J anomaly is weak and discontinuous at the latitude of the Flemish Cap-Galicia sections. Red lines represent seismic profiles used in this study. Orange: Landward limit of oceanic crust. Yellow: Oceanward limit of continental crust. BB Bay of Biscay, GB Galicia Bank, IAP Iberia Abyssal Plain, MAR Mid-Atlantic Ridge, NNB northern Newfoundland Basin, NS Newfoundland Seamounts, SNB southern Newfoundland Basin, TAP Tagus Abyssal Plain, ODP Ocean Drilling Program, DSDP Deep Sea Drilling Program, COB Continent ocean boundary.

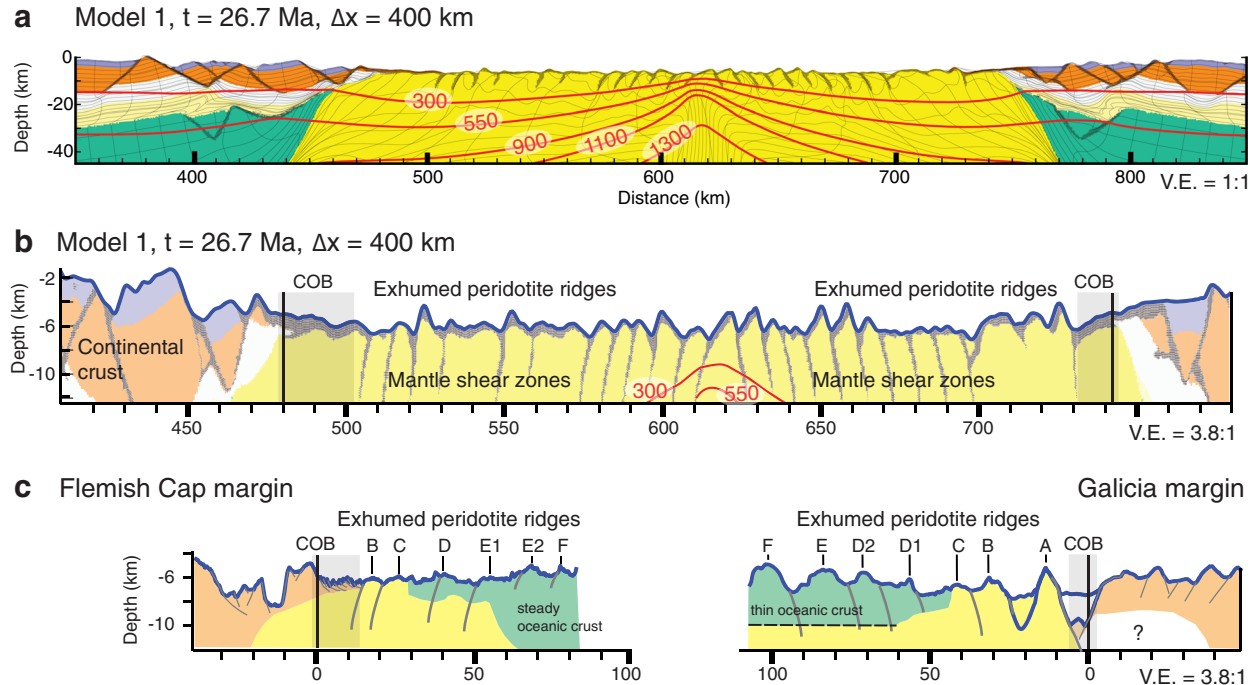

**Fig. 2 Thermo-mechanical model of mantle exhumation during magma-poor rifted margin formation and Flemish Cap-Galicia conjugate margin geometry. a** Numerical model results (shown for a sub-region of the model domain, see Supplement Movies S1 and S2 and Fig. S7 for full-time evolution). $t$ = time since onset of extension, $\Delta x$ = extension at uniform velocity 1.5 cm/yr. Contours of isotherms in degrees Celsius. Shown are upper crust (orange), middle crust (white), lower crust (light yellow), pre-rift sediments (purple), lithospheric mantle (green), sub-lithospheric mantle (yellow), overlay of weakened frictional-plastic shear zones (gray). **b** Topography of the basement from the numerical model with vertical exaggeration. We note that the density structure of this model is not calibrated to fit the relative elevation between continents and mid-ocean ridge resulting in over-estimated depth. Colors: same as in **a**. **c** Topography of the top basement offshore along SCREECH-1 and WE-1/ISE-1 seismic profiles with the same vertical exaggeration. Gray: interpreted shear zones responsible for the successive formation of basement highs named here with a letter.

margins (e.g., ISE-1/WE-1)[7,25,26] (Fig. 1), and sampling of peridotite rocks[27] reveal the following key characteristics of the transitional domain. (1) Syn-rift sediment deposition is very low to absent. (2) A narrow peridotite ridge (ridge A, Fig. 1a) bordering the edge of the last continental crust along the Iberian margin, with highly serpentinized mantle rocks, limited partial melting, and moderately steep flanks dipping 20–28°[1,27,28], characterized by (3) fast syn-rift exhumation from upper-mantle depths (150 km) around 120 Ma as revealed by Pressure-Temperature-time (PTt) data from the ridge[29–32]. (4) A proximal domain of magma-poor exhumed mantle on both the Galicia and Flemish Cap conjugate margins with rough and angular basement topography and asymmetric ridges with amplitudes of 500–1600 m between the base and the top of the structures (ridges B, C, and D1 on Fig. 1a and B, C, D on Fig. 1c). (5) A distal transitional domain on the Galicia side with evidence of episodic magmatic activity and discontinuous stretches of thin oceanic crust with smoother basement topography, and (6) several basement highs with amplitudes of 1000–2300 m (Galicia: ridges D2, E, and F on Fig. 1a). (7) A distal transitional domain on Flemish Cap profile showing typical seismic velocities characteristic of 5–6 km thick oceanic crust, interpreted as the start of magmatic spreading between highs E1 and E2 (Fig. 1c)[6,24]. (8) On average 700–800 m shallower basement topography in the distal transitional domain as compared to the proximal (Fig. 1 and supplemental Fig. S1). In this work, we use 2-D thermo-mechanical geodynamic modeling of continental rifting and mantle exhumation and integrate this with a new analysis and kinematic reconstruction of the transitional domain from continental to oceanic crust on the magma-poor Galicia-Flemish Cap conjugate margins. We find that multiple out-of-sequence

detachments with recurring dip reversal form during magma-poor rifting and mantle exhumation as a consequence of the strong competition between weak frictional-plastic shear zones and the thermally weakened necking domain beneath the exhuming footwall. The forward model reproduces the observed characteristics comprising elevations (1–2 km) and wavelength of exhumed mantle highs (10–15 km) for rift velocities of 15 mm/yr and the formation of the first peridotite ridge and inferred thickness of the brittle layer. Predictions from the forward model and the new kinematic reconstruction of the exhumed mantle domain along the Galicia-Flemish Cap conjugate margins provide time constraints on crustal breakup (121.0 ± 1.0 Ma) and inception of a mature spreading center (113.8 ± 1.6 Ma).

## Results

**Modeling results**. We use an arbitrary Eulerian–Lagrangian finite-element method[33] to model continental extension of a layered lithosphere with frictional-plastic and thermally activated power-law viscous rheologies at upper-mantle scale and high resolution (see supplementary material, Tables S1 and S2, Fig. S4 and S5). Strain localization in the model is a consequence of frictional-plastic weakening with accumulating strain (Fig. S4). Reference model 1 shown here (Fig. 2 and supplement) builds on earlier forward models of lithosphere extension[34] and represents one among many we have investigated (Fig. 3 and supplement). The forward model is characterized by intermediate crustal strength rheology (wet quartz) that allows for a moderate decoupling between the upper-crust and lithospheric mantle. Key features not included in earlier models are increasing the thermal conductivity of mantle rocks with decreasing temperature leading to more efficient heat transfer by diffusion close to the

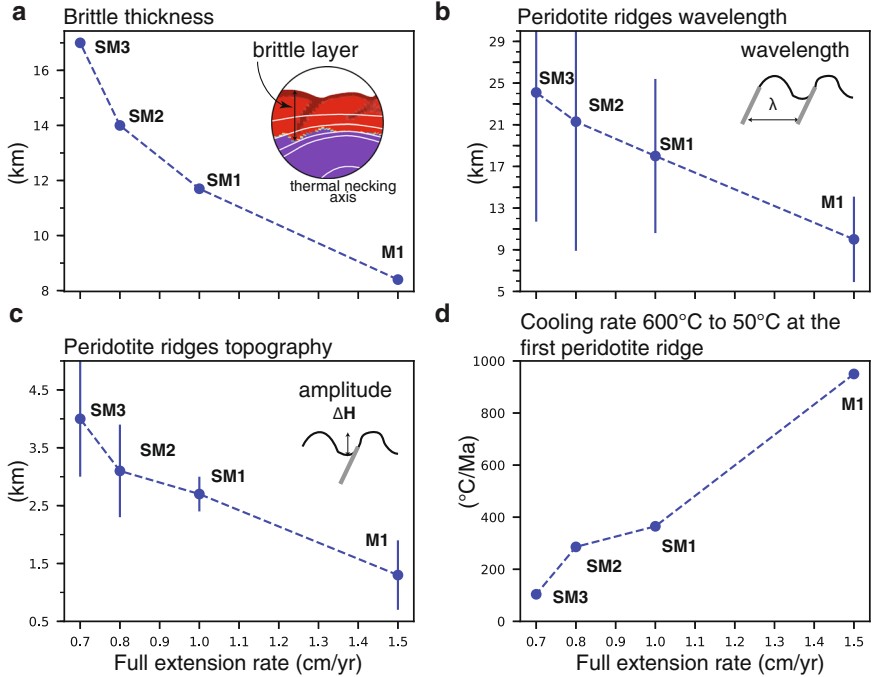

**Fig. 3 Sensitivity of model behavior to varying spreading rates. a** Brittle thickness on top of the active detachment fault root zone next to the thermal necking axis. Large-scale view of the plastic (brittle) domain in model M1 can be found in supplemental Fig. S8. **b** Average wavelength and standard deviation of the peridotite ridges. **c** Average amplitude and standard deviation of the exhumed peridotite ridges. **d** The cooling rate of the exhumed mantle at the first peridotite ridge. Each point corresponds to one model run (Snapshots and model animations SM1-3 are available in the supplement material). These models only differ from model M1 by their spreading rate. We note that all these models exhibit symmetric spreading (see supplementary Fig. S9 and S10).

surface[35,36], and separation of viscous and frictional-plastic strain allowing for strain localization of exhuming sub-lithospheric mantle (see supplementary material for details, Fig. S6). The final stage of the forward model after lithospheric breakup and a total of 400 km of continuous extension comprises a ~225 km wide exhumed mantle domain characterized by fault-bounded peridotite ridges (Fig. 2a, b). Deformation follows three phases: (1) upper-crustal and upper-mantle normal faulting and viscous necking during about 7 Myr, (2) distal margin formation during ~5 Myr leading to crustal breakup at 12 Myr, and (3) mantle exhumation to the seafloor (see supplementary information and animation for details on full model evolution). Phase 1 and 2 are well documented in previous studies[34,37] (see supplementary Fig. S7). We note that we did not aim to reproduce the detailed style of faulting during the formation of the distal continental margin with oceanward dipping faults rooting in the S detachment[9]. Here, we focus specifically on the characteristics of phase 3 during which the mantle is exhumed to the seafloor by large offset normal faults. The normal faults form in sequence at a high angle within the frictional-plastic part of the exhuming mantle on top of the thermal necking zone, which is characterized by a brittle thickness of ~8 km in agreement with estimates for the Flemish Cap-Galicia conjugates[21] (Fig. 3 and supplementary Fig. S8). Following exhumation, the fault surfaces rotate and form low-angle exhumed detachment surfaces with domal relief and asymmetric footwall scarps with on average alternating dip orientation through time (Fig. 4a and Supplementary Movies S1 and S2). Detachment faults are active for ~1.5 Ma. The maximum offset of the detachment faults is in the order of 15 km after which a new fault forms in the thermal necking zone of the footwall. During the first stage of strain migration in the thermal necking zone, initial strain localizes on two conjugate faults resulting in an uplifted fault-bounded footwall scarp until one fault takes over and forms a new detachment fault. At the same time, the fault root zone migrates away with the hanging wall from the thermal

necking zone until it is deactivated leaving a landward dipping shear zone. Successive formation of single asymmetric detachments results in symmetric distribution of large offset landward dipping brittle shear zones on either side of the thermal necking zone. Domes and footwall scarps are elevated up to 2 km relative to the surrounding seafloor and exhibit a wavelength of ~9–10 km (Figs. 2 and 3).

The degree of strain weakening and the thickness of the frictional-plastic layer of the exhuming mantle domain provide main controls on the offset and distribution of asymmetric large offset detachments. Higher strain weakening ratios than used here results in a highly asymmetric distribution of the exhumed mantle domain around the necking zone not consistent with the largely symmetric distribution of exhumed mantle observed along the Galicia-Flemish Cap conjugate margin system (see supplementary Fig. S9, S11, and S12). Low weakening limits the offset of the normal faults and does not reproduce large offset asymmetric detachments as observed (e.g., see supplementary Fig. S13). The limited thickness of the brittle layer in the exhumed mantle domain, here in the order of 7–8 km for a spreading rate of 1.5 cm/yr (Fig. 3 and supplementary Fig. S8), favors flexural footwall rotation and large offset faults[15]. Strain migrates to form a new asymmetric detachment above the thermal upwelling with alternating dip orientation, as the active normal fault migrates away from the necking zone (Fig. 4a). Varying extension rates between 0.7 and 1.5 cm/yr results in brittle layer thickness varying between 17 and 8 km (Fig. 3). The main characteristics that change with extension rate and brittle thickness are the magnitude of footwall topography, characteristic wavelength, and cooling rate during exhumation. Lower spreading rates imply a lower cooling rate and thicker brittle layer resulting in symmetric a-magmatic spreading characterized by larger wavelength and higher amplitude exhumed mantle highs (Fig. 3). We note that fault types and detachment styles can locally vary at lower spreading rates with for instance formation of domino

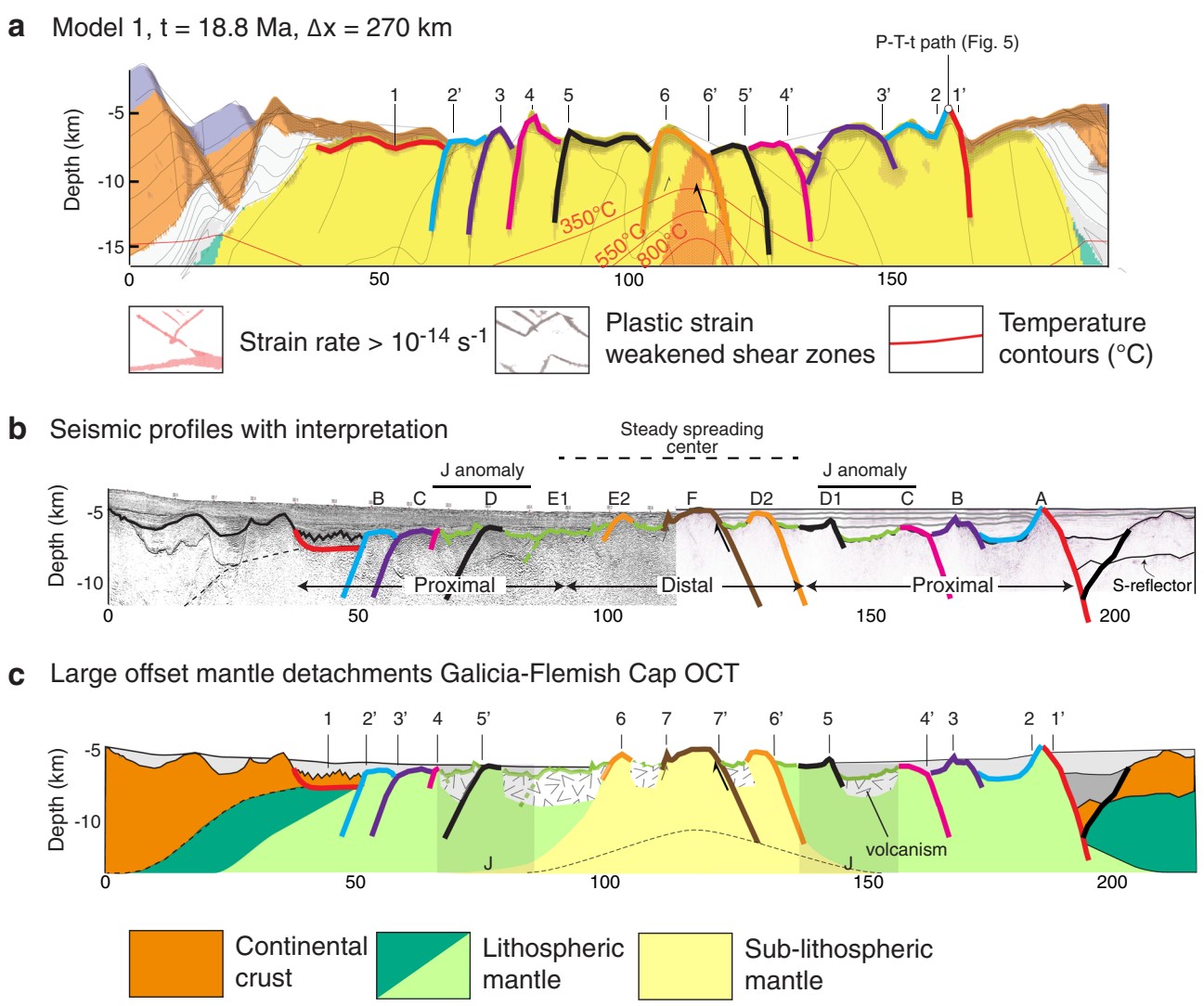

**Fig. 4 Forward model prediction of the exhumed mantle, observed transitional domain of the Flemish Cap-Galicia conjugate margins with the interpretation of detachment faults in order of formation. a** Model configuration at 18.8 Ma. Large offset faults are numbered in sequence of formation linking footwall scarps around the fault breakaway (numbers without a prime) with their detachment fault root zones (numbers with a prime). **b** Interpretation of the transitional magma-poor domain on top of the seismic profiles in terms of conjugate footwall scarps and their detachment fault root zones (colored) and magmatic additions (green). West of dome E2, Hopper et al.[6] argue for the presence of a continuous 2 km oceanic crust (Fig. 2c). The domain between highs C and E2 exhibits a rough high-frequency top basement topography with a small amplitude that we interpret as small offset high-angle faults associated with volcanism. The top basement morphology on Galicia side with similar characteristics is also interpreted as magmatic addition. The position of the J anomaly is located at the transition from incipient volcanism to a steady spreading center. **c** Same interpretation of the transitional magma-poor domain in terms of conjugate footwall scarps (numbers without a prime) and their detachment fault root zones (numbers with a prime) and nature of the transition zone. Incipient volcanism starts after detachment 4. Note that interpretation of the nature of the ultra-distal small blocks as remnants of continental crust here on top of detachment 1 on Flemish Cap side is similar to Sutra et al.[49] but that the kinematic we infer for their formation differs from their interpretation.

short-offset high-angle normal faults and related rotated mantle blocks sitting on top of a long-offset detachment fault (see supplementary Fig. S10). Models without the increase of thermal conductivity with low temperature at shallow depths result in lower topographic highs and smaller wavelength for a given extension rate owing to slower cooling and a thinner brittle layer (see supplementary Fig. S15).

**Model implications for the Galicia-Flemish Cap system.** Our model results reproduce the first-order characteristics of the transitional domain of the Galicia-Flemish Cap conjugate margin system (Fig. 2a–c). The basement morphology of the proximal first 45 km of the transitional domain is asymmetric with a first

peridotite ridge adjacent to the last continental crust and domes on the Galicia side and a low relief region on the Flemish Cap side comparable to our model. Spacing between highs and domes of ~10–15 km and their elevation between 700 m and 2300 m are similar to first order. The elevation and the symmetric steep flanks of the peridotite ridge are also similar to the first mantle high generated in our model. The model configuration at 18.8 Ma after 270 km of extension (Fig. 4a) shows the formation of six detachments with corresponding footwall highs and domal relief. Each large offset detachment leaves a footwall scarp on one conjugate and a corresponding detachment root zone on the other conjugate, where the corresponding footwall scarp and detachment root zone are numbered in order of their formation (Fig. 4a). Basement morphology of the transitional domain on the

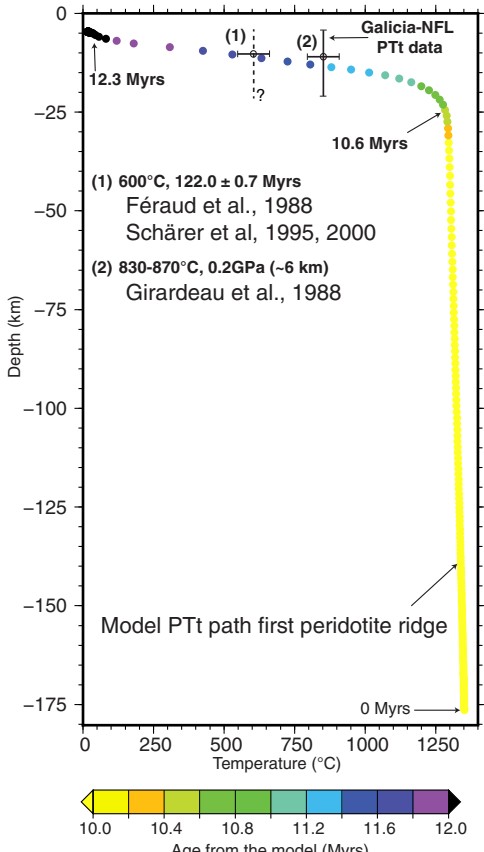

**Fig. 5 Pressure-temperature-time (PTt) path of the first peridotite ridge in the numerical model.** The mantle exposed on top of the first peridotite ridge in the numerical model (see location in Fig. 4a) is at $t = 0$ Ma located at 175 km depth. The PTt path follows the adiabatic gradient before the mantle is being cooled down quickly during the last stage of exhumation from 1300 °C at 25 km depth to ~50 °C at 5 km depth in <2 Myr. This PTt path is in agreement with observations made of the exhumed peridotite ridge along the Galician margin sampled by ODP leg 103, hole 637a[29–32]. Observed mylonitization during cooling between 1000 and 860 °C, serpentinization during brittle deformation, and low cogenetic partial melting are compatible with the predicted PTt path[29].

conjugate Galicia-Flemish Cap margins is similarly interpreted allowing to link these basement features on both conjugates in order of formation from the proximal to the distal part of the transitional domain (Fig. 4b, c). Although episodic magmatic activity is not included in the forward model it reproduces and explains the first-order features of this transition zone following crustal breakup and before the establishment of a stable mid-oceanic spreading center.

The Pressure-Temperature-time (PTt) history of the first peridotite ridge associated with crustal breakup in our model (Fig. 4a, ridge 1) indicates extremely fast (>600 °C/Ma) cooling and exhumation from temperatures of 600 °C to the surface within ~1 Myr (Figs. 3 and 5). Dating of the peridotite ridge at site 637 gives a consensus age of ~122.0 ± 0.7 Ma with a closure temperature ~600 °C[30–32] (see Supplementary Table S3). Using the cooling rate predicted by the model suggests that crustal breakup and exhumation of the peridotite ridge to the surface in the northern segment of the Iberia-Newfoundland conjugate margins occurred ~121.0 ± 1.0 Ma. The slightly younger age suggested by Ar/Ar dating on syn-tectonic plagioclase (117.7 ± 0.9 Ma) with a closure temperature ~200 °C[38], may be a consequence of uncertainties associated with modeling of diffusion

kinetics of argon during extremely fast cooling[39] or late fluid-induced recrystallization that causes Ar loss[40,41]. The rate of mantle upwelling and the cooling rate during mantle exhumation are directly controlled by the extension rate (Fig. 3). The Ar/Ar age on plagioclase of 117 Myr would indicate a very low extension velocity of ~0.5 cm/yr that is not consistent with the inferred rate of ~1.5 cm/yr for this time interval[42]. This apparent contradiction may result from uncertainties with Ar/Ar plagioclase dating. We note that the M0 magnetic anomaly, which is dated at 121.4 Ma with a duration of 0.4 Myr, is close to the crustal breakup time inferred here. Other older M-magnetic anomalies pre-date crustal breakup implying that these anomalies should not be visible on the seafloor at this latitude and therefore cannot be used to constrain kinematic reconstructions. These anomalies might, however, be visible towards the south as a consequence of syn-rift northward propagation-related diachronicity of the deformation in the Iberia-Newfoundland system[43–45].

Our results can also be used to infer the time of inception of a mature steady-state mid-ocean ridge spreading center provided constraints on average spreading rate and distance to the first oceanic crust. The average spreading rate between crustal breakup at 121 Ma and the first undisputed magnetic anomaly C34n dated at 83 Ma[46] can be estimated to 1.6 cm/yr, given the distance from the most distal location of continental crust to C34n, 305 km on the Flemish Cap side versus 310 km on Galicia. We note that this estimate is close to and consistent with the spreading rate of 1.5 cm/yr used in our reference model. On the Flemish Cap conjugate margin the first well-defined oceanic crust is recognized at 60 ± 5 km from the COB based on seismic refraction velocities[6,24], which implies an age of 113.8 ± 1.6 Ma for the start of a mature magmatic spreading center assuming a constant average spreading rate of 1.6 cm/yr between 121 and 83 Ma, and that the seismic line is parallel to the flowline. On the Galician margin, the symmetry of the system implies that ridge D1, which is at ~60 km distance from the most distal location of continental crust, represents the start of a mature magmatic spreading center on the conjugate. We interpret the distal transitional domain on both sides (Figs. 1a, c and 4) as resulting from slow mature mid-ocean ridge spreading characterized by along-strike variations in magmatic output and locally formation of detachments faults[3–5]. Both bathymetry and top basement morphology suggest magmatic addition in the distal domain. On average 700–800 m shallower basement can be explained with the isostatic consequence of the addition of 4 km thick oceanic crust in the distal domain, while post-rift sediment loading explains ~100 m elevation difference (see supplementary Fig. S1–3). The distal Flemish Cap transitional domain exhibits high-frequency top basement topography with small amplitude that we interpret as small offset high-angle faults associated with magmatic spreading (Fig. 4b). The distal Galicia transitional domain exhibits domes with larger amplitude and wavelength compared with the proximal domain. This morphology suggests a segment where only a part of the extension is accommodated by magmatic injection and that the remaining low extension rate is accommodated by detachment faults with larger amplitudes and wavelength, as observed at slow and ultra-slow mature mid-ocean ridges[3–5]. The age of 113.8 ± 1.6 Ma for the start of a mature magmatic spreading center corresponds well to the so-called Albian-Aptian breakup unconformity[47]. The J anomaly appears spatially correlated with the earliest start of mid-ocean ridge spreading which suggests that these regional events are associated closely in time[46]. However, we note that this correlation is not proof of causality since the J anomaly is a magnetic anomaly that might be polygenic and is not continuous along the distal transitional domain of the Newfoundland-Iberia system[48] (Fig. 1e).

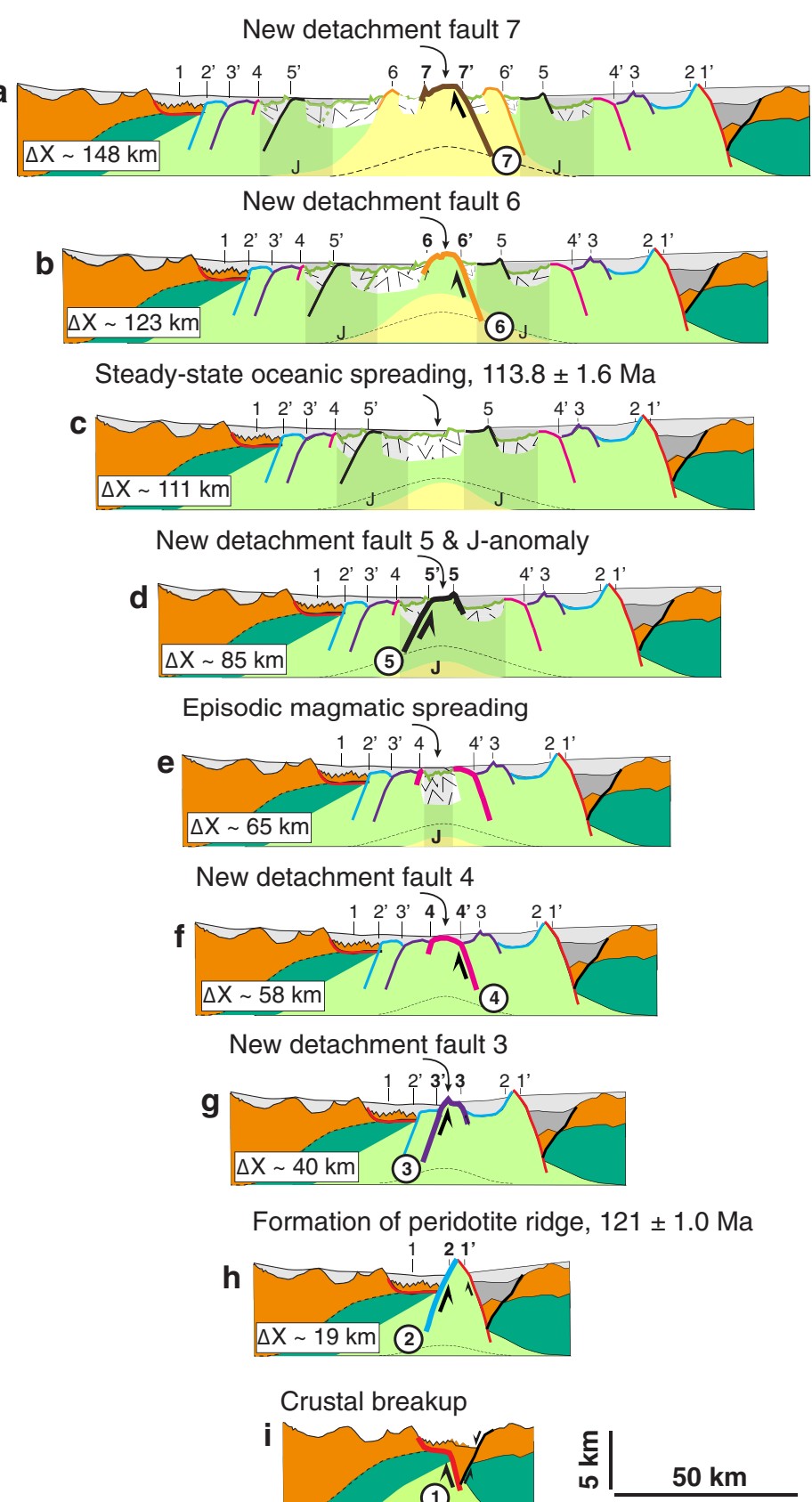

**Fig. 6 Kinematic reconstruction of the conjugate seismic profiles SCREECH-1 and ISE-1/WE-1. a**, **b** Final stage of mantle exhumation with the formation of detachment fault 6 and 7 during mature mid-ocean ridge spreading. **c** Onset of magmatic spreading with the accretion of thin oceanic crust with smooth basement morphology. **d** Formation of detachment fault 5. **e** Episodic magmatic activity resulting in the accretion of thin oceanic crust and smooth basement morphology and formation of J anomaly. **f**, **h** Successive formation of detachment faults 2–4. **i** crustal breakup and formation of first peridotite ridge at 121 Ma.

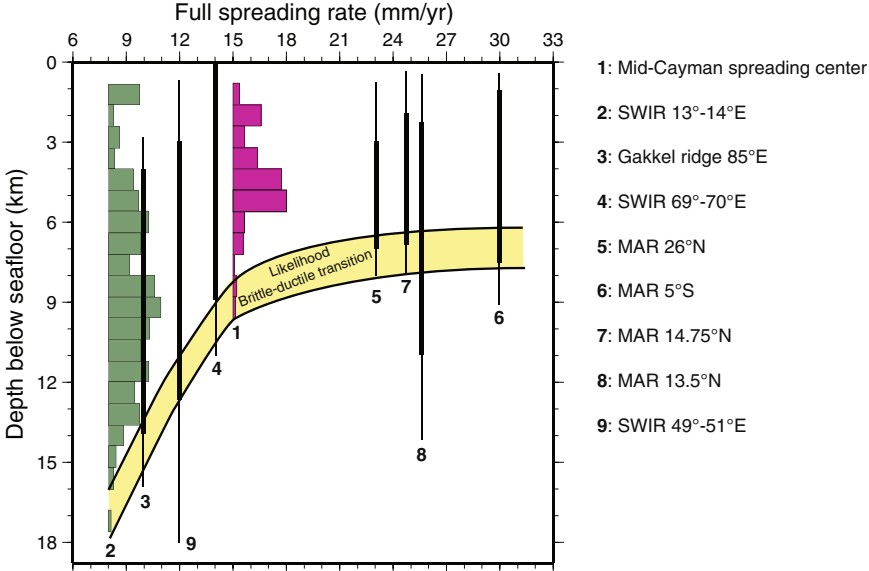

**Fig. 7 Observed brittle thickness at mid-ocean ridges as inferred by the micro-seismicity distribution with depth for various spreading rates.** Depth is given using the seafloor elevation as a reference. Histograms show the vertical distribution of micro-seismicity with depth[51]. Bold lines show the depth distribution of the majority of micro-earthquakes while adjacent thin lines represent isolated micro-earthquakes[53,65–70]. **1**, **2**: Grevemeyer et al., 2019[51]; **3**: Korger and Schlindwein, 2014[68]; **4**: Katsumata et al., 2001[53]; **5**: deMartin et al., 2007[65]; **6**: Tilmann et al., 2004[66]; **7**: Grevemeyer et al., 2013[67]; **8**: Parnell-Turner et al., 2017[69]; **9**: Yu et al., 2018[70]. Modified from Grevemeyer et al., 2019[51].

We employ the forward model to guide a kinematic reconstruction of the transitional domain by making two basic assumptions (Fig. 6). (1) We assume that each basement high can be interpreted as a footwall scarp that has a corresponding fault root zone on the conjugate margin during symmetric exhumation, which is characteristic of model behavior (Fig. 4b, c). (2) We assume that the conjugate margin system formed symmetrically consistent with the distribution of transitional domain on the conjugate margins and use the inferred average spreading velocity. With these assumptions, the distance from the most distal location of continental crust equates to the time after a crustal breakup. This reconstruction is by definition non-unique as absolute ages for these characteristic basement features do not exist but builds on the insight in a-magmatic mantle exhumation from the forward model (Fig. 6). The implied temporal evolution allows reconstructing the conjugate margin system to each stage of large offset normal fault formation (Fig. 6). We infer the following stages for the formation of the transitional domain going back in time. (1) Final stage of the transition zone with the formation of detachment fault 6 and 7 during mature mid-ocean ridge spreading (Fig. 6a, b). (2) Onset of mature magmatic spreading with the accretion of thin oceanic crust with smooth basement morphology (Fig. 6c). (3) Formation of detachment fault 5 (Fig. 6d). (4) Episodic magmatic activity resulting in the accretion of thin oceanic crust and smooth basement morphology and formation of J anomaly (Fig. 6e). (5) Successive formation of detachment faults 2–4 (Fig. 6f–h). (6) Crustal breakup and formation of the first peridotite ridge at 121 Ma (Fig. 6h, i). We note that detachment 1 post-dates the formation of the sub-crustal S-detachment (Fig. 4b) that requires a root zone beneath the Flemish Cap margin[6,9,49]. The origin of the formation of the small continental blocks at the ultra-distal Flemish Cap margin proposed here (Fig. 6h, i) differs from the previous interpretations[6,49]. Assuming on average 15 km displacement on each of the seven large offset normal faults and a total width of the magma-poor mantle domain of ~140 km suggests that ~75% of the total amount of extension was accommodated by a-magmatic spreading, and 25% by intermittent magmatic

activity. We acknowledge that this is an average and that magmatic spreading is likely more important in the distal than in the proximal transitional domain.

## Discussion

The model results presented here are relevant for and can be directly compared to slow and ultra-slow mid-ocean ridge spreading systems such as for instance along the South-West Indian Ridge (SWIR) and the Cayman spreading center that exhibit spreading rates 7–15 mm/yr similar to those inferred here for the formation of the transitional domain along the Galicia-Flemish Cap conjugate margins[50–53]. Values of brittle thickness from models presented here are comparable with estimates based on the distribution of micro-seismicity at active slow and ultra-slow spreading mid-ocean ridges (Fig. 7). Brittle layer thickness along the eastern segment of the SWIR (69–70°E) and the Cayman system that is spreading at ~15 mm/yr[51,53], is at least 7–9 km as in our reference model, while the ultra-slow oblique segment of the SWIR (13–14°E), which is spreading at lower rates down to ~8 mm/yr, is characterized by a brittle layer thickness of up to 16 km[51] (Fig. 7). The two ultra-slow segments of the SWIR at 62°E and 64°E, spreading at 12–14 mm/yr, are dominated by detachment faults that form in a "flip-flop" mode of faulting[11–13] consistent with the models presented here. The exhumed mantle highs along these two segments have average elevations 1.4 ± 0.7 and 1.8 ± 0.4 km and wavelength 16.9 ± 5.6 and 19.2 ± 3.0 km, respectively, higher than those observed along the Galicia-Flemish Cap margin. We suggest that these differences can be attributed to a lower spreading rate resulting in a higher thickness of the brittle layer following the trend observed in our model results (Fig. 3). Similarly, domino short-offset high-angle normal faults and related rotated mantle blocks sitting on top of a large offset detachment fault interpreted on the proximal Antarctica–Australia exhumed mantle domain[18,54] are comparable to model behavior at an ultra-slow spreading rate (see supplementary Fig. S10). The results presented here demonstrate that these characteristic features can be explained by alternating weak large offset brittle normal faults without recourse to more complex rheologies[12,18,21].

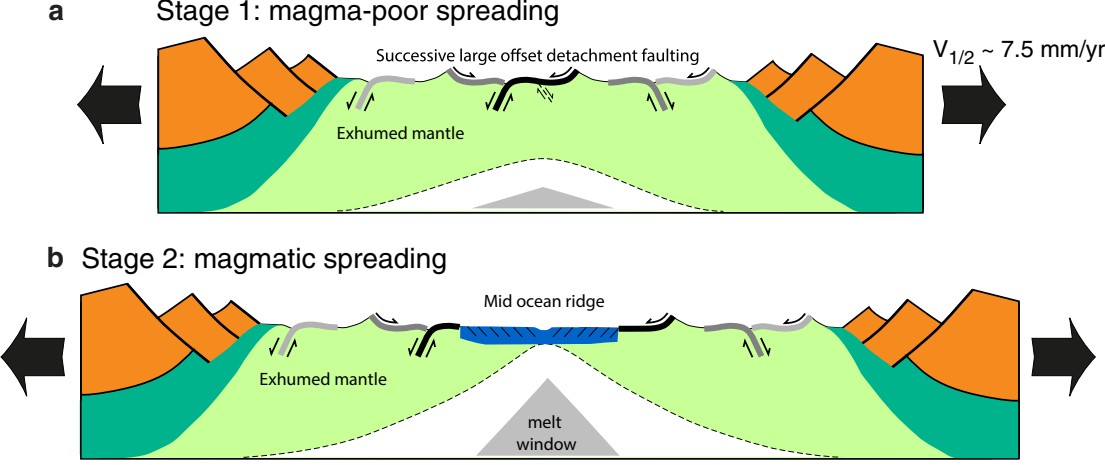

**Fig. 8 Template showing main characteristics from magma-poor spreading to magmatic spreading on Galicia-Flemish Cap margins. a** Phase of mantle exhumation with the successive formation of large offset detachment faults with recurrent dip reversal. **b** Magmatic spreading with dominant accretion of normal oceanic crust and small offset normal faults and locally formation of detachments faults and exhumed mantle domes for ultra-slow and slow spreading rates.

The results presented here show that the formation of exhumed mantle domes along magma-poor margins and slow to ultra-slow mid-oceanic spreading systems is controlled by the successive formation of large offset normal fault zones with on average opposing dip, with new fault zones forming above the thermal necking zone rupturing the exhumed fault surface and consequently leaving genetically linked root zones and detachment footwall scarps on either side of the conjugate margin and/or spreading center (Fig. 8a) consistent with earlier conceptual models[4,8,10,13,14,18]. This genetic link is used here to establish a new interpretation of the transitional magma-poor domain along the Galicia-Flemish Cap conjugate margins indicating that seven large offset normal faults and intermittent minor episodes of magmatic addition have controlled its formation dominated by a-magmatic spreading. The new interpretation allows reconstructing the kinematic evolution of the transitional domain, the time of continental crustal breakup, and the time of inception of mature mid-ocean ridge spreading for these conjugate magma-poor margins. Magma-poor spreading is discontinued once sufficient mantle melt allows the establishment of a mature magmatic spreading center characterized by accretion of normal thickness oceanic crust and locally exhumed mantle domes (Fig. 8b). The numerical models and new reconstruction presented here provide a new framework for understanding and interpreting magma-poor margins and ultra-slow mid-ocean ridge spreading systems.

## Methods

**Seismic data.** The two seismic profiles from Flemish Cap and Galicia come from published studies, respectively, by Hopper et al.[6] and Dean et al.[26]. The first one results from Kirchoff prestack depth migration on data acquired aboard R/V Maurice Ewing and R/V Oceanus in 2000. The second one results from post-stack Kirchoff Time Migration followed by depth conversion based on smoothed interval velocities on data acquired by the *RV Marcus G. Langseth* in 2013.

**Thermo-mechanical model.** The forward numerical model of rifted margin formation and mantle exhumation is conducted using the highly efficient finite-element code Fantom[33,34], which employs an arbitrary Lagrangian–Eulerian discretization together with the material point method to solve for Stokes flow:

$$\nabla\left(\mu_{\text{eff}}\left(\nabla\mathbf{v} + (\nabla\mathbf{v})^T\right)\right) - \nabla P + \rho g = 0$$

where $P$ is the pressure, $\mathbf{v}$ is the velocity, $\mu_{\text{eff}}$ is the effective viscosity, $\rho$ is the density, and g is the gravity acceleration. This is coupled with time ($t$)-dependent

heat conservation

$$\rho c_p \frac{DT}{Dt} = \nabla(k\nabla T) + H + v_z \alpha_T g T \rho$$

where $T$ is the temperature, $c_p$ the specific heat capacity, $\alpha_T$ thermal expansion, $H$ the radiogenic heat production, and the last term is the temperature correction for adiabatic heating and cooling when material moves vertically at velocity $v_z$. This coupling is done through nonlinear temperature ($T$) and pressure ($P$) dependent rheologies, as well as the temperature dependence of buoyancy

$$\rho = \rho_0\left(1 - \alpha_T\left(T - T_0\right)\right)$$

where $\rho_0$ is the reference density at $T_0$, temperature at surface conditions, using the Boussinesq approximation, which considers that changes in density ($\rho$) are small enough to approximate the conservation of mass by an incompressible flow.

$$\nabla \cdot \mathbf{v} = 0$$

**Model setup.** The model domain is 1200 km long in the horizontal direction ($x$) and 600 km in depth ($z$). It is discretized using 2400 × 290 finite elements, leading to spatial resolution of 500 m along $x$ and 200 m along $z$ in the top 20 km, 625 m between 20 and 70 km depth, 1100 m between 70 and 125 km depth, and circa 8000 m in the sub-lithospheric mantle (bottom 475 km) (Fig. S2). The initial geometry includes four horizontal layers: upper crust (25 km), lower crust (10 km), mantle lithosphere (90 km), and the sub-lithospheric mantle (475 km), with reference densities $\rho_0$ set to 2750, 2900, 3300, and 3300 kg m$^{-3}$ at $T_0 = 0$ °C. The coefficient of thermal expansion, $\alpha_T$ is constant and equal to $3.1 \times 10^{-5}$ °C$^{-1}$ for all phases. Boundary conditions include extensional velocities applied to the lithosphere at $v_{x1/2} = \pm 0.75$ cm yr$^{-1}$ on the left and right sides and the corresponding exit flux is balanced by a low-velocity inflow in the sub-lithospheric mantle. The sides and base are vertical and horizontal free slip boundaries, respectively. The lithosphere is characterized by a centered 400 km wide mechanical heterogeneity represented by white noise in the initial strain field. This approach is designed to represent inheritance from previous tectonic phases. A small thermal heterogeneity is introduced at the base of the lithosphere in the model center in order to enhance rift localization (Fig. S2 and S3).

**Rheology.** When the state of stress is below the frictional-plastic yield stress, the flow is viscous and is specified by temperature-dependent nonlinear power-law rheologies based on laboratory measurements on "wet" quartz[55] and "wet" olivine[56]. The effective viscosity, $\mu_{\text{eff}}$, in the power-law rheology is of the general form:

$$\mu_{\text{eff}} = f A^{-1/n} \dot{E}_2^{(1-n)/2n} e^{\frac{Q+Vp}{nRT}}$$

where $\dot{E}_2$ is the second invariant of the deviatoric strain rate tensor $\frac{1}{2}\left(\dot{\varepsilon}_{ij}\dot{\varepsilon}_{ij}\right)$, $n$ is the power-law exponent, $A$ is the pre-exponential scaling factor, $Q$ is the activation energy, $V$ is the activation volume, $p$ is the pressure, $T$ is the absolute temperature in Kelvin, and $R$ is the universal gas constant. $A$, $n$, $Q$, and $V$ are derived from laboratory experiments and the parameter values are listed in Table S1. The factor $f$ is used to scale viscosities calculated from the reference quartz and olivine flow laws. This scaling produces strong and weak crust and reproduces the difference between "wet" and "dry" olivine.

Frictional-plastic (Mohr-Coulomb) yielding occurs when:

$$\sigma_y = \sqrt{J_2} = C\cos(\phi_{eff}) + P\sin(\phi_{eff})$$

where $J_2$ is the second invariant of the deviatoric stress, $C$ is the cohesion, $\phi_{eff}$ is the effective internal angle of friction following $P \cdot \sin(\phi_{eff}) = (P - P_f)\sin(\phi)$ and $P_f$ is the pore-fluid pressure. This approximates frictional sliding in rocks, including pore-fluid pressure effects. The model accounts for frictional-plastic strain softening, which is responsible for strain localization in the brittle layer[57]. Weakening of faults and shear zones may be caused by a range of mechanisms including accumulating strain responsible for cohesion loss, high transient (short term) or static fluid overpressures, mineral transformations as for instance serpentinization or formation of phyllosilicates responsible for a reduction of the friction angle. Strain weakening is introduced by a linear decrease of $\phi_{eff}(\varepsilon)$ from 15° to 2° and $C(\varepsilon)$ from 20 to 4 MPa with respect to plastic strain ($\varepsilon$) between $\varepsilon_0$ and $\varepsilon_1$[15,57]. In our model $\varepsilon_0$ and $\varepsilon_1$ are set to 5% and 105%, respectively. In the model presented here, we use a lower weakening ratio for exhuming mantle rocks $\phi_{eff}(\varepsilon)$ from 15° to 4° and a constant cohesion of 20 MPa. Higher weakening ratios result in a highly asymmetric exhumed mantle domain with shear zones with offsets far exceeding 15 km and therefore incompatible with observations. Rate and amplitude of the reduction of the friction angle and cohesion are calibrated to reproduce wavelength of fault migration and flexural elevations building on previous work[15,57]. Shear heating and viscous weakening and/or hardening are not included in our models. Applying such processes would require a new bulk frictional parameterization to reproduce the strength balance and the pattern of fault migration but would not modify the conclusions of our study.

**Fault strength**. In the models presented here, we have focused on calibrating the degree of strain weakening in the mantle to reproduce the observed structures in the transitional magma-poor domain along the Galicia-Flemish Cap conjugate margins (see supplementary Figs. S9–15). In order to reproduce the observed symmetry of this domain between the two conjugate margins, the wavelength, and the topography of peridotites ridges we have needed to limit the degree of strain weakening in the mantle materials to generate symmetric spreading and out-of-sequence strain migration in the footwall during mantle exhumation. We note that the weakening parameters of the continental crust are based on previous modeling work and may benefit from further calibration. Adopting lower weakening for the crust would result in slightly different fault network geometry but not change the main insights gained here on mantle exhumation. We also note the study of Hansen et al.[58] that demonstrates significant viscous strain hardening following grain size reduction for olivine at conditions characteristic for the brittle-ductile transition. While we have not included this additional complexity, it may provide an explanation for reduced effective strain weakening in mantle materials. Alternatively higher fault healing as suggested by Püthe and Gerya[16] or different fault dynamics at a short time scale (velocity-strengthening of talc gouge[59]) may explain reduced effective strain weakening in mantle materials.

**Thermal model setup**. The initial geotherm (Fig. S2) corresponds to a steady-state geotherm with a fixed surface and Moho temperatures, respectively, of 0 °C and 550 °C, heat production of 1.1 and 0.5 µW m$^{-3}$, respectively, for the upper and lower crust and a constant thermal diffusivity of $10^{-6}$ m$^2$ s$^{-1}$ except for the mantle at low temperatures. To account for the increasing diffusivity of olivine with lower temperature the thermal conductivity of mantle rocks increases by a factor of 2.5 between 876 °C and 0 °C[35,36] (Fig. S4). Mantle potential temperature is 1280 °C. The bottom temperature in the model domain is 1520 °C. The initial temperature of the base of the lithosphere is 1328 °C at 120 km depth. The sub-lithospheric mantle follows an adiabatic temperature gradient of 0.4 °C/Km. To avoid secular cooling of the lithosphere and to maintain heat flux to the base of the lithosphere the thermal conductivity linearly increases to 51.46 W/m/K from 1330 to 1350 °C (~125 km depth), which corresponds to scaling the thermal conductivity by the Nusselt number of upper-mantle convection[34]. The initial temperature is laterally uniform except at the LAB where an initial small thermal anomaly is included in the center of the model to seed lithospheric necking (Fig. S2 and S3). The model does not include melt prediction. We believe this appropriate given that formation of the transitional magma-poor domain along the Iberia-Newfoundland conjugate margins is mostly by a-magmatic extension.

**Neglecting magma supply**. Various mechanisms have been proposed to explain the magma-poor nature of the transitional domain including (but not limited to) low mantle potential temperature[60], ultra-slow spreading rate, and counter-flow of depleted lower lithospheric mantle[61,62]. As the degree of magmatism is very low to absent, we have chosen to ignore mantle melting in our modeling approach. We believe that counter-flow of the inherited depleted mantle is the most viable hypothesis to explain the absence of mantle melting as there is evidence for the depleted mantle in the transitional magma-poor domain from IODP/ODP sampling and dredging[10,63], as there is no evidence for an anomalous low mantle temperature[60], and as inferred extension rates for the final phase of rifting and mantle exhumation are in the order of 1.5 cm/yr full spreading rate sufficient for decompression melting. In our models, we have chosen for simplicity not to include counter-flow of depleted lower lithospheric mantle and have opted instead

to disable mantle melting. A large part of the exhumed mantle domain in our models can be interpreted as representing a depleted mantle that does not produce decompression melting. We similarly interpret the magma-poor nature of the exhumed mantle on the Galicia-Flemish Cap conjugate margins (Figs. 4c and 6). The limited magmatic spreading during the formation of the ultra-slow transitional domain (<25% on average) inferred in this study supports the idea that neglecting magma supply is a valid approximation and mantle melting is not a critical missing factor.

## Data availability
All data are available in the main text or supplementary materials.

## Code availability
Numerical models are computed with published methods and codes, described in the Methods section and supplementary materials.

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

## Acknowledgements

This study is funded through the COLORS project by TOTALENERGIES (RH) and The Research Council of Norway. We thank Uninett Sigma2 for computing the time of project NN4704K (RH).

## Author contributions

T.T. contributed the numerical models and analyzed the data for the Iberia-Newfoundland margins. R.H. assisted with the 2-D modeling and contributed ideas on rifted margin styles. Both authors contributed to developing the concepts and to writing the manuscript.

## Competing interests

The authors declare no competing interests.
