## [Peer Review File · Nature Communications]

REVIEWER COMMENTS

Reviewer #1 (Remarks to the Author):

The manuscript entitled "Mantle exhumation at magma-poor rifted margins: a competition between frictional shear zones and thermally weakened necking domains", written by Theunissen and Huisman presents a forward geodynamic model that reproduce the tectonic deformation observed at magma poor continent-ocean transition (COT). Based on seismic observations multiple authors suggest that exhumed mantle domain at COT formed by flip-flop sequential detachment faults (Gillard, Autin, et al., 2016; Gillard, Manatschal, et al., 2016; Reston & McDermott, 2011), however up to now no geodynamic models were able to reproduce this observation. By adding two physical properties in an already published 2D thermo-mechanical model the authors are able to mimic in morphology and in size of the observed structures. The topic of the study is crucial to understand the onset of a new seafloor-spreading ridge and will interest the tectonic community. The paper is well written and illustrated even if some figures could be improved (see detailed comments). I raised four main points and made multiple detailed comments, which should be discussed prior to publication:

1) Definition of oceanic crust, proximal/distal transitional domain: In the text as well as on the figures there is no clear definition of the structural characteristics of the different domains (thinned continental crust, proximal transitional domain, distal transitional domain, oceanic crust). This is critical to follow the comparison between the seismic and the model. Watching several times the movie, I think that the model convinces me that the seafloor spreading starts at ~22Myr, when the flip flop detachment systems are shorter and more symmetric (Do you agree with that?). However, on the Iberia-Newfoundland I do not clearly see the shortening of the detachment systems I only see a top basement which is 700 to 800 m shallower compared to the proximal exhumed mantle. I'm interpreting the distal part of the two lines as proto-oceanic (Gillard et al., 2019) and suggest that the steady state seafloor spreading is further oceanward. The definition of a steady state formed oceanic crust is very difficult here as the classical well layered 6-7 km thick oceanic crust is absent from the seismic profiles, so your definition of oceanic crust should be clearly explained in the text.

2) Clearly differentiate the geological interpretation from the geodynamic model and the kinematic reconstruction: In the section model implications(...) it is hard to follow what is extracted from the model and what is observed. I suggest to double check this section to clearly state from km to km we model And we observe similar structures on the seismic sections from km to km.

3) Emphasized more the physical reason, which permit to reproduce an already observed architecture of the domain. The observation of sequential faulting has already been made but the novelty resides in the implementation of new parameters (modification of mantle thermal conductivity and the separation of viscous/frictional strain). I think that this aspect should be more highlighted and discussed, could you for instance provide a model without those properties?

4) The first part of the discussion (L179-193) is repetitive. This section is a summary; It would have some interest to discuss what are the limits of the modelling strategy and what could be added (e.g. volcanic addition).

Detailed comments

L8; a-magmatic: What is a a-magmatic mantle? Is the term a-magmatic needed? Indeed, in most of the cases exhumed serpentinized mantle at ocean-continent transition is associated with volcanic and magmatic rocks.

L8-L11: the sentence is long and could be divided in two. I rather suggest to write a first sentence highlighting the complex architecture at COT and then state that the mechanism causing this complexity are poorly understood.

L17; Predictions... a new interpretation: The interpretation of flip flop detachment was first made based on seismic observation (Gillard, Autin, et al., 2016; Reston & McDermott, 2011). I think that the modelling can

rather find some potential mechanical causes for this observed process.

L18 Galicia-Newfoundland: If you want to be specific to the two conjugate lines, you can use Galicia-Flemish Cap conjugate margin and use Iberia-Newfoundland for the wider system. Also name could be added on the map Fig.1

L23: What is the definition of a late mature mid-ocean ridge spreading?

L25-26 well-defined...conjugates margins: On the Iberia Newfoundland margin the first well defined magnetic anomaly is C34 and it is located hundreds km away from the first steady state oceanic crust (Nirrengarten et al., 2017). However, I agree that the size of the serpentinized domain are more or less symmetric on the conjugate sides. I suggest to modify the sentence.

L26: The proximal part of the transition: State somewhere before that the zone of exhumed mantle is the transition zone in your study.

L40-43: This sentence is just a summary of what has been exposed before.

L45-46: Add that you focus on the northern Iberia-Newfoundland conjugate margins, because the timing of deformation is diachronous along strike (Mohn et al., 2015; Nirrengarten et al., 2018)

L48-49: highly serpentinized... melting: add that it is based on seismic velocity interpretation

L47-58: Could you add the different described domains on the Fig 1

L53: with elevations 500-1600 m above the seafloor: Is it an average seafloor or the deepest seafloor or the variation of topography between the base and the top of the structure?

L54-58: Is the distal exhumed mantle domain formed by seafloor spreading? Where is the clear oceanic crust on the sections?

L59; new analysis and interpretation: The interpretation of multiple detachment was already made no?

L79-80: Could you add the duration of each phases extract from the model or refer to Fig. S5

L84-85: Could you estimate the thickness of the frictional-plastic part of the model? And hence tell the rooting depth of the normal faults. Can it be compared to the depth approximated in other studies (Gillard et al., 2019)?

L126: from proximal to distal margin: I think that it is from the proximal to the distal transition zone no? Could you show the limit between the two domain on Fig.3

L130-139: In this three sentences you assume that the modelling is perfectly adapted to the Iberia-Newfoundland case and that the age of closure at 600° is more robust than the Ar-Ar dating of syn-tectonic plagioclase. I wonder if the model is by itself sufficient to state this.

L139-142: The regional...breakup: Could the regional unconformity be linked to the initiation of steady state seafloor spreading? I do not understand well why this sentence is here.

L149: is recognized at 60 km: What are the criterion to determine well defined oceanic crust on Screech 1.

L150-151: with an age ..between 121-83 Ma: You should also assume that the line is oriented parallel to the flowline.

L152-153: What are the characteristics indicating the presence of magmatic additions?

L156-157: The J-anomaly is a magnetic anomaly which is poly-genic and poly-phased (Nirrengarten et al., 2017) on the northern part of the conjugate margin this anomaly is less pronounced then farther south and the position of the first steady state oceanic crust is different between different authors (Dean et al., 2015; Gillard, Manatschal, et al., 2016; Nirrengarten et al., 2017; Sutra et al., 2013). At one point in the document it would be nice to see your criterion to determine steady state oceanic crust.

L 180 non volcanic: prior it was magma poor choose one terminology.

L186-188: The structural interpretation has already been published (Dean et al., 2015; Gillard, Manatschal, et al., 2016), the novelty of the paper is rather the determination of physical processes permitting to reproduce the interpreted architecture (increasing thermal conductivity of mantle rocks with decreasing temperature, and separation of viscous / frictional plastic strain).

L188-190: already said before

Figure1: -The J anomaly is not punctual et rather occupies a wider zone (Whitmarsh & Miles, 1995)

- On the figure the transitional domain is called Zone of Exhumed Mantle but on the legend panel it is written may be exhumed mantle, oceanic crust, continental crust. I suggest writing exhumed mantle with potential magmatic addition or remnants of continental crust

- Fig 1E Colors from the legend panel do not match the one on the map. ODP-DSDP sites instead of IODP moreover they are not plotted on the map.

Figure 2b: In the presented model at the active spreading system, the top basement is around 5km below seafloor, isn't it to deep? Could you comment on that

-L251 pre-rift sediments (purple)

Figure 3: How Figure 3B has been drawn? What are the reasons to add volcanism?

-Detachment 6 do not have the same color on a and b sub-figures.

- I suggest to invert sub-figure b and c to have the most interpretative section at the bottom.

- Light green exhumed mantle could you rather show the serpentized mantle because in a way all the mantle is exhumed from deeper level.

-Add the color code on the figure

Fig S1: I think that the observed top basement (blue curve) should be corrected by flexural back-stripping.

Indeed the current topography has encompassed sediments loading and thermal subsidence, in order to compare the model basement topography and the picked top basement you should perform a back-stripping (Nirrengarten et al., 2016).

References

- Dean, S. L., Sawyer, D. S., & Morgan, J. K. (2015). Galicia Bank ocean-continent transition zone: New seismic reflection constraints. *Earth and Planetary Science Letters*, 413, 197–207. <https://doi.org/10.1016/j.epsl.2014.12.045>
- Gillard, M., Manatschal, G., & Autin, J. (2016). How can asymmetric detachment faults generate symmetric Ocean Continent Transitions? *Terra Nova*, 28(1), 27–34. <https://doi.org/10.1111/ter.12183>
- Gillard, M., Autin, J., & Manatschal, G. (2016). Fault systems at hyper-extended rifted margins and embryonic oceanic crust: Structural style, evolution and relation to magma. *Marine and Petroleum Geology*, 76, 51–67. <https://doi.org/10.1016/j.marpetgeo.2016.05.013>
- Gillard, M., Tugend, J., Müntener, O., Manatschal, G., Karner, G. D., Autin, J., ... Ulrich, M. (2019). The role of serpentization and magmatism in the formation of decoupling interfaces at magma-poor rifted margins. *Earth-Science Reviews*, 196(June), 102882. <https://doi.org/10.1016/j.earscirev.2019.102882>
- Mohn, G., Karner, G. D., Manatschal, G., & Johnson, C. A. (2015). Structural and stratigraphic evolution of the Iberia-Newfoundland hyper-extended rifted margin: a quantitative modelling approach. *Geological Society, London, Special Publications*, 15, 13810. <https://doi.org/10.1144/SP413.9>
- Nirrengarten, M., Manatschal, G., Yuan, X. P., Kuszniir, N. J., & Maillot, B. (2016). Application of the critical Coulomb wedge theory to hyper-extended, magma-poor rifted margins. *Earth and Planetary Science Letters*, 442. <https://doi.org/10.1016/j.epsl.2016.03.004>
- Nirrengarten, M., Manatschal, G., Tugend, J., Kuszniir, N. J., & Sauter, D. (2017). Nature and origin of the J-magnetic anomaly offshore Iberia–Newfoundland: implications for plate reconstructions. *Terra Nova*, 29(1). <https://doi.org/10.1111/ter.12240>
- Nirrengarten, M., Manatschal, G., Tugend, J., Kuszniir, N., & Sauter, D. (2018). Kinematic Evolution of the Southern North Atlantic: Implications for the Formation of Hyperextended Rift Systems. *Tectonics*, 37(1), 89–118. <https://doi.org/10.1002/2017TC004495>
- Reston, T. J., & McDermott, K. G. (2011). Successive detachment faults and mantle unroofing at magma-poor rifted margins. *Geology*, 39(11), 1071–1074. <https://doi.org/10.1130/G32428.1>
- Sutra, E., Manatschal, G., Mohn, G., & Unternehr, P. (2013). Quantification and restoration of extensional deformation along the Western Iberia and Newfoundland rifted margins. *Geochemistry, Geophysics, Geosystems*, 14(8), 2575–2597. <https://doi.org/10.1002/ggge.20135>
- Whitmarsh, R. B., & Miles, P. R. (1995). Models of the Development of the West Iberia Rifted Continental-Margin at 40-Degrees-30n Deduced from Surface and Deep-Tow Magnetic-Anomalies. *Journal of Geophysical Research-Solid Earth*, 100(B3), 3789–3806. <https://doi.org/10.1029/94jb02877>

Reviewer #2 (Remarks to the Author):

Review of the manuscript NCOMMS-21-34634, Mantle exhumation at magma-poor rifted margins: a competition between frictional shear zones and thermally weakened necking domains, by Thomas Theunissen and Ritske S. Huismans

The MS submitted by T. Theunissen and R. Huisman investigates the mechanisms and structures (mantle ridges, detachments) that develop during mantle exhumation in the transition zone from continental to oceanic crust of distal magma-poor rifted margins. The authors present the results of forward geodynamic numerical models and provide explanations for the geometry and origin of exhumed mantle peridotite ridges. The authors propose that during extension, strength competition between weak frictional-plastic shear zones and thermally weakened necking domain develops, favouring the formation of multiple out-of-sequence detachments with recurring dip reversal, permitting emplacement of mantle ridges as a result of cross cut between successive detachments.

The authors then use the first order characteristics of their numerical results to guide kinematic reconstruction of the transitional domain offshore the system of Newfoundland-Galicia conjugated margins pair, and to propose a model of emplacement of peridotite ridges and magma, also providing time constraints on the last stages of breakup and first oceanic crust emplacement at these margins. Specifically, the model highlights that remnants of the crosscutting detachments are now located in the exhumed mantle domain of both sides of the conjugated margins pair. The model also supports an Albian-Aptian initiation of a mature magmatic spreading centre.

The authors finally propose that their model also provides explanations for emplacement of similar structures at the South-West Indian Ridge and the Cayman spreading centre, suggesting that their numerical model can be applied to other magma-poor margins and magma poor spreading centres, that strongly broadens the significance of the study.

I have a good knowledge of the structures and nature of the rock materials at the Galicia-Newfoundland conjugated margins from geophysical and sampling data, but I am not a numerical modeller and my knowledge of this approach is based on work published by other researchers.

I find, the MS very well written and of very good quality with clear explanations of the context and appropriately citing references. The work is well described and should be possible to reproduce by others. The presented data and figures, both in main text and in supplementary, are suitable to introduce the first order tectonic structures observed at the studied margins and to support the analytical approach and discussion of the MS. The numerical model and associated parameters look robust and valid to me: the proposed model, in my opinion, convincingly reproduce the first order structures of the mantle domain at the Galicia-Newfoundland margins, as presented in fig 1. Predicted age of exhumation of the first PR on the Galicia side is right in the range of estimated age resulting from IODP sampling, and the predicted age of 113.8 ± 1.6 Ma for the start of a mature magmatic spreading center is consistent with the generally agreed Albian-Aptian age of breakup, further supporting the proposed numerical model. The proposed explanations for the mechanisms and kinematics of emplacement of the structures observed at the distal domain of the Galicia-Newfoundland margins appears robust and consistent with existing data and knowledge at this margin, as well as with structural model proposed for the emplacement of crosscutting detachments at oceanic spreading centre (e.g. Reston, 2018), thus strongly supporting the conclusions of the MS.

I have, however, two main questions regarding 1) the effect of uniform (1.5 cm/yr) extension velocity used in the models on the formed structures, and 2) on the change of faulting regime from in-sequence faulting during crustal hyper-extension to out-of-sequence during mantle exhumation, please see details below (main questions highlighted in bold). I also suggest below minor editions of the main text and figures and ask some more minor questions.

I hope my feedback will be useful and will help reviewing and publishing this MS.

Wishing to the authors good luck in the next stages of editions of the MS.

Best regards,
Gaël Lymer

Main text:

L. 53: "asymmetric ridges with elevations 500-1600 m above the seafloor"

This suggests to me that all ridges outcrop above the seafloor, but not all the ridges actually reach the seafloor, and only 2 on Fig. 1. Elevations laterally change a lot along the ridges in 3D. Maybe precise the sentence?

L.55: "and (5) several basement highs between 1000-2300 m elevated above the seafloor"

It is not clear to me where this is, is it in the oceanic domain? Maybe precise which basement you refer to

and/or point them on figure 1?

L.84-97: I think it would help the reader if the description of the model provided in this section of the main text was supported by visual markers on the figures (Fig. 2, Fig. S5) pointing to the features described in the text.

L.151: "On the Galician margin ridge 3a is at approximately 60 km distance from the most distal location of continental crust"

I cannot find ridge 3a. Is it ridge D1 on fig.2? Or ridge 5 on Fig. 3? Please check.

L.153: "On average 700-800 shallower basement can be explained with the isostatic consequence of 6 km thick oceanic crust (Fig. S1)."

Can volume-increase magmatic additions also contribute to shallow this domain?

L. 207: "Our models provide therefore a framework for understanding formation of both the exhumed mantle along non-volcanic rifted margins and for the a-magmatic segments of ultra-slow mid-ocean ridge systems."

Reston 2018 also provided explanations for mid-ocean ridge systems in a sensibly similar way, although based on structural analysis rather than numerical simulation. Maybe also refer to his work at this place? (currently only cited line 37 if I am correct).

Figures:

Fig. 2 (Also Figure S5, Table S2 and supplement movies S1 & S2):

Main question 1: In the presented model, the extension occurred at uniform velocity of 1.5 cm/yr. Other studies have predicted an acceleration of extension before continental break-up, with a transition from slow to fast extension during hyper extension, development of detachment and mantle exhumation (Brune et al. 2016, Abrupt plate accelerations shape rifted continental margins, Nature volume 536, pages 201–204). As rift evolution appears strain-rate-dependent, I wonder what would be the implication in your models of a uniform vs accelerating velocity of extension on the development of structures in the distal margins? Have you made models with varying velocities?

Main question 2: The structures with fault-bounded peridotite ridges obtained in the final stage of the forward model strongly resemble the wide zone of exhumed mantle observed from seismic data at the Galicia margin. It is also consistent with proposed models for tectonic process of accretion of oceanic crust at mid-oceanic ridges (e.g. Reston, 2018).

Although the authors clearly specify that the focus of this paper is the exhumed mantle domain, differences exist between the presented model and the observed structures of the hyper-extended domain of the Galicia margin at the edge of the continental crust. There, at least in the area of location of line WE-1 shown in Fig. 1, the block bounding faults are systematically dipping toward the ocean over >30 km above the S detachment (e.g. Lymer et al. 2019). This does not seem to be simulated in the presented model.

The authors refer to previous publications for the development of the necking domain and distal margin (L. 79), but I think the paper should be self-consistent and I wonder if the change of faulting regime from synthetic faults in the hyper-extended domain above S detachment, to out-of-sequence faulting in the exhumed mantle domain, resulting in flip-flop detachments, should be discussed or at least mentioned in this paper? Also, could this change be related to the aforementioned variations of velocity of extension?

Fig. 3 & Fig. 5: Please highlight Galicia side and Newfoundland side.

A confusing point to me here is that the S detachment on the Galicia side is not indicated. Instead on figs 3 & 5, detachment 1 (red (unofficial colour for S in different publications)) appears to underline continental crust on the Newfoundland side (Fig. 3b), as does the S on the Galicia side. On figs 3&5, detachment 1 under continental crust is shown oceanward of the COB, while no continental crust is shown oceanward of the COB on Fig1C,D (although the key clearly says the transitional domain can have various nature). Figs 3 (km 200) & 5 currently display a black fault on the Galicia side, which is not mentioned in the text.

In summary: Detachment 1 currently looks to me like the S detachment on the Galicia side, making figures 3&5 looking like if they were in the wrong sense (i.e. Galicia on the left).

I think these should be clarified; some ideas: highlight Galicia/Newfoundland; change the colour of the material above detachment 1 on the NWFDL side from crust to mantle or undefined? Show the S on the Galicia side? define the black fault?

Fig. S1: Can you precise how you obtained the basement plot (blue curve), is it interpretation from figure 2

or from a reference?

Fig. S3: The inherited weak domain instead of a single seed, allowing more freedom for fault development is a great idea! Such "more freedom" parameter was missing from other publications I have read on numerical models.

Tables S1 & S2: Except my point on uniform velocity of extension (see comment for fig. 2), the bounding conditions of table 2 and mechanical and thermal conditions of table 1 look good to me, although I am no specialist on this matter.

Reviewer #3 (Remarks to the Author):

Review to Mantle exhumation at magma-poor rifted margins: a competition between frictional shear zones and thermally activated necking domains by T. Theunissen and R.S. Huismans

I have read the paper and supplementary material carefully and I must say that it is difficult to evaluate the modelling part of the manuscript, as I was asked to, because there is only one finely tuned model presented and it is impossible to evaluate how the spacing and timing of the flip-flop depends on softening, rate of spreading etc.. As the timing is critical for the kinematic reconstruction at the end, it seems to me that the authors should demonstrate they have a robust model and they need to display how the change in the parameters affects the solution.

In the current form, I really don't see what "your model add to current understanding of non volcanic margin and ultra slow mid-oceanic ridge system" as you conclude. I agree that the model (you could call M instead of M1 as there is only one model) produces flip-flops... but the flip flop conceptual model is established for a long time, and the parallel between magma poor margins and slow spreading ridge is in my opinion textbook material (master level) today but I might be biased by my proximity to M. Cannat and S. Leroy. The big questions are how to produce rolling hinge down to 16 km depth not 7 or 10 km (e.g. Bickert et al. 2020), and how it relates to melt. Any simulations with softening of the mantle material and no melt can produce flip-flop (see per exemple Jourdon et al 2019 : 5 lines in a paper "flip-flop are produced consistent to observations at Galicia margin and SWIR") . Taras Gerya family models generally don't produce the flip flop because they are all coupled with melt as he really intend to model oceanic crust.

When I accepted to review the paper and I saw the authors which I respect, I was really expecting a complete parametric study to be present in the supplementary material so that the flip-flop behavior in model without melt could be considered as a nailed thing that we do not need to model anymore. The only thing I found was "the main changes between model M1 as compare to other published models are the conductivity of the mantle and some fine tuning of the softening parameters". How these two parameters affect the prediction of the simulations ? Well this is key if you want to say that modelling has brought something to our understanding of Galicia margin because you need to make sure your timing between flipflop are robust to build your kinematic model and attribute what is missing to the melt that is absent of your simulations.

Also a parametric study would have been interesting in order to better assess if the model obtained explains how to form detachment that are brittle down to 14km depth like observed at ultra-slow spreading ridge or if there is a need for new ductile laws like proposed by the modelling study of Bickert et al. 2020. In other words, if the same simulation is run with a slower rate like 7mm/yr of extension does it produce deeper detachment? (I would argue yes, maybe because in Jourdon et al. they seem to be deeper and the extension rate is slower) but still this is not in this paper while there is a figure that show how they deepens with rate in nature in the supplementary material.

To me it seems this paper has been redirected straight from a shorter format nature family journal because the authors cannot really make their point in a short format. So I would recommend the paper for major revision to give the author the time to actually move some of the modelling material like very good figure S6

to be in the main text but also to actually push the parametric study in the suppI. Material and partly in the main text to strengthen their new timing arguments about the Galicia margin. I am sure that the authors, who are serious scientists, did perform this parametric study for them-selves so it should not be a lot of extra work and I think it is important that the reader and the reviewer can assess how the best fitting model is sensitive to variation of new parameters and rate of extension. Below I also list some minor comments.

With best regards
Laetitia Le Pourhiet

===== minor comments =====

I don't understand why the crustal material soften more than the mantle... I would have expected the opposite as the mantle go through serpentinisation while the continental crust does not produce phyllosilicates as weak as talc and serpentines. How does this choice affect your result ?

How the lack of melt and its feedback on deformation in the formulation of the model affects the result. I can see from the snapshots that the isotherm 1300°C is at 30-35 km depth from 12-16 Ma on. In these conditions the mantle would probably melt (and you do propose it does on the cartoon of Fig 6) so it would be good to actually at least post process the melt produced by the model and discuss how its presence would affect the dynamic of the flip-flop. Moreover computing the actual timing of melt would be of great help to calibrate your simulation to the Galicia margin and strengthen your arguments about kinematic reconstruction.

The model has no erosion sedimentation could you discuss how sedimentation would affect the relief observed on the margin. I expect they could grow larger with sediment infill.

At line 137, you suggest that your model is correct and that Argon dating is be wrong. There are many ways for your model not to be correct, including thermal blanketing by sediments which would delay cooling. I would just say that your model does not capture the slow cooling suggested by the ArAr age but that this could be due either to some approximation in your model or to the issue of diffusion in Ar Ar dating.

At line 139 : your interpretation of regional unconformity is not based on your modelling result since you do not simulate melt. Please write this interpretation is not based on your model in the text.

Paragraphe | 144 to 157 : I really don't see how your model result participate to reach your conclusion... all your arguments are based on kinematics and data.

Paragraph | 159 to 176...

I am not sure I understand well the reasoning and if I do, I think this paragraph should come earlier in the paper and it should be make clearer to the reader.

Basically, do you assume :

1/ that your model without magma provides a robust timing (this should be proved by a complete parametric study that shows that the timing between to detachment faults is independent of the softening parameters and the rate of extension when you get the spacing correct) and that thanks to your model one can just count the detachment to get back to time.

2/ that when the detachment are not spaced like in you numerical model then the rest of extension is magmatic addition (without modelling them or at least predicting from the TM when, where and how much melt is produced)

With the result you present here you cannot claim that your model also explains the ultra slow segment of the SWIR with detachment down to 16 km depth, because you do not show any model with flip flop that root

at 16 km and you know as well as I know that it is very difficult to operate a rolling hinge exhumation in a layer with such a plastic thickness.

Instead of suggesting that the differences are due to the rate, please provide a model with a lower rate of extension and actually demonstrate it by including the parametric study you must have run to get the perfect model 1. Compare also your approach to the paper of Bickert et al. which does capture the 16 km depth and the spacing but need to include complex plastic behavior based on actual thin sections of sample dragged at the ridge to manage to root the decollement as deep.

REVIEWER COMMENTS

Reviewer #1 (Remarks to the Author):

The manuscript entitled “Mantle exhumation at magma-poor rifted margins: a competition between frictional shear zones and thermally weakened necking domains”, written by Theunissen and Huisman presents a forward geodynamic model that reproduce the tectonic deformation observed at magma poor continent-ocean transition (COT). Based on seismic observations multiple authors suggest that exhumed mantle domain at COT formed by flip-flop sequential detachment faults (Gillard, Autin, et al., 2016; Gillard, Manatschal, et al., 2016; Reston & McDermott, 2011), however up to now no geodynamic models were able to reproduce this observation. By adding two physical properties in an already published 2D thermo-mechanical model the authors are able to mimic in morphology and in size of the observed structures. The topic of the study is crucial to understand the onset of a new seafloor-spreading ridge and will interest the tectonic community. The paper is well written and illustrated even if some figures could be improved (see detailed comments). I raised four main points and made multiple detailed comments, which should be discussed prior to publication:

We appreciate the very detailed and constructive comments of the reviewer. We sometimes refer to a previous or a next answer to avoid repetition.

1

1) Definition of oceanic crust, proximal/distal transitional domain: In the text as well as on the figures there is no clear definition of the structural characteristics of the different domains (thinned continental crust, proximal transitional domain, distal transitional domain, oceanic crust). This is critical to follow the comparison between the seismic and the model.

We have modified the text and the figures to better describe the structural characteristics of the different domains (See also comments 11, 12 and 13).

2

Watching several time the movie, I think that the model convince me that the seafloor spreading starts at ~22Myr, when the flip flop detachment systems are shorter and more symmetric (Do you agree with that?).

We should clarify that the model does not include formation of a mature magmatic mid-ocean ridge spreading system including mantle melting and oceanic crust. The full model evolution exhibits a-magmatic spreading leading to mantle exhumation.

We have clarified the model setup description in the main text and also improved the description of the model results. In the model, a-magmatic mantle exhumation starts just after crustal breakup at 12 Myr in the model. There is neither melt prediction nor melt extraction in our model and it is therefore not possible to define the start of a mature mid-ocean ridge spreading system. We agree that the flip-flop mode of faulting becomes more regular at 22 Myr compared to previous stage between 12 Myr and 22 Myr. It takes time to establish a steady-state thermal necking at the spreading center. This transient behavior explains the small difference in spacing and the more regular mode of flip-flop detachments after 22 Myr.

3

However, on the Iberia-Newfoundland I do not clearly see the shortening of the detachment systems. I only see a top basement which is 700 to 800 shallower compared to the proximal exhumed mantle. I'm interpreting the distal part of the two lines as proto-oceanic (Gillard et al., 2019) and suggest that the steady state seafloor spreading is further ocean ward. The definition of a steady state formed oceanic crust is very difficult here as the classical well layered 6-7 km thick oceanic crust is absent from the seismic profiles, so your definition of oceanic crust should be clearly explained in the text.

We agree that it is challenging to define where mature magmatic spreading starts on these conjugate sections. The magmatic oceanic crust along the Flemish Cap profile starting between domes E1 and E2 shows typical characteristics of 5-6 km thick oceanic crust and has been interpreted as the start of a magmatic spreading center (Funck et al., 2003; Hopper et al., 2004). The corresponding distal domain on the Galicia conjugate margin shows 2-3 domes (D2, E and F on figure 4b) and evidence for episodic magmatic addition. We interpret this distal domain as resulting from slow mature mid-ocean ridge spreading characterized by along strike variations in magmatic output and episodic formation of detachments faults (e.g., Escartin et al., 2008; Howell et al., 2019). This is the reason why we prefer to characterize this distal domain as the start of a mature MOR system.

We now better describe our interpretation in the text on these domains.

4

2) Clearly differentiate the geological interpretation from the geodynamic model and the kinematic reconstruction: In the section model implications(...) it is hard to follow what is extracted from the model and what is observed. I suggest to double check this section to clearly state from km to km we model And we observe similar structures on the seismic sections from km to km.

We have more clearly separated the interpretation of the model and the kinematic reconstruction in the text. We provide ridge labels and positions in kilometers as defined in the model result or in the observations to clearly distinguish model and observations.

5

3) Emphasized more the physical reason, which permit to reproduce an already observed architecture of the domain. The observation of sequential faulting has already been made but the novelty resides in the implementation of new parameters (modification of mantle thermal conductivity and the separation of viscous/frictional strain). I think that this aspect should be more highlighted and discussed, could you for instance provide a model without those properties?

Thank you for this comment. We agree and we have now included a full set of models exploring the sensitivity to strain weakening, full extension rate, and thermal conductivity in the supplement (as also suggested by reviewer 3). We have modified the discussion to better highlight this important aspect.

6

4) The first part of the discussion (L179-193) is repetitive. This section is a summary; It would have some interest to discuss what are the limit of the modelling strategy and what could be added (e.g. volcanic addition).

We agree. We have rewritten the discussion to better highlight the physical reasons of model behavior (Comment 5) and to provide model limitations (as also suggested by reviewer 3).

7

Detailed comments

L8; a-magmatic: What is a a-magmatic mantle? Is the term a-magmatic needed? Indeed, in most of the case exhumed serpentized mantle at ocean-continent transition is associated with volcanic and magmatic rocks.

Volcanics are sparse in this exhumed mantle domain. However we agree that there is some evidence of small magmatic addition in the exhumed mantle domain (Girardeau et al., 1988; Evans and Girardeau, 1988, Funck et al., 2003; Hopper et al., 2004). We now use “magma-poor”.

8

L8-L11: the sentence is long and could be divided in two. I rather suggest to write a first sentence highlighting the complex architecture at COT and then state that the mechanism causing this complexity are poorly understood.

We agree and we have updated this sentence accordingly.

“The transition zone from continental crust to the mature mid-ocean ridge spreading centre of the

Iberia-Newfoundland magma-poor rifted margins is mostly composed of exhumed mantle characterized by highs and domes with varying elevation, spacing and shape. The mechanism controlling strain localization and fault migration explaining the geometry of these peridotite ridges is poorly understood.”

9

L17; Predictions... a new interpretation: The interpretation of flip flop detachment was first made based on seismic observation (Gillard, Autin, et al., 2016; Reston & McDermott, 2011). I think that the modelling can rather find some potential mechanical causes for this observed process.

We agree and we have modified the last part of the abstract.

“ Predictions from the forward model are used to guide a new kinematic reconstruction of the exhumed mantle domain along the Galicia-Flemish Cap conjugate margins providing time constraints on crustal breakup and inception of a mature spreading center. Model behaviour also shows that fault types and detachments styles vary with spreading rate and fault strength and confirm that these results can be directly compared to other magma poor passive margins such as along Antarctica-Australia and to ultra-slow mid-ocean spreading systems as the South-West Indian Ridge.”

10

L18 Galicia-Newfoundland: If you want to be specific to the two conjugate lines, you can use Galicia-Flemish Cap conjugate margin and use Iberia-Newfoundland for the wider system. Also name could be added on the map Fig.1

This is a good suggestion that we have followed in the entire manuscript now.

11

L23: What is the definition of a late mature mid-ocean ridge spreading?

We have modified this sentence and provide the definition in the text.

12

L25-26 well-defined...conjugates margins: On the Iberia Newfoundland margin the first well defined magnetic anomaly is C34 and it is located hundreds km away from the first steady state oceanic crust (Nirrengarten et al., 2017). However, I agree that the size of the serpentinized domain are more or less symmetric on the conjugate sides. I suggest to modify the sentence.

We agree with this comment. Thank you. We have simplified this sentence.

13

L26: The proximal part of the transition: State somewhere before that the zone of exhumed mantle is the transition zone in your study.

We have better defined the different domains at the start of the introduction (see also comments n°1, 11 and 12) and we have modified this sentence.

“ The distal continental margin adjacent to the **transitional** domain involves crustal detachment faulting resulting from the coupling of crustal thinning and mantle exhumation during the last stage of continental rifting.”

14

L40-43: This sentence is just a summary of what has been exposed before.

We have modified this sentence.

“The mechanism controlling the mode of faulting, strain migration during mantle exhumation, and formation of peridotite ridges such as along the Flemish Cap-Galicia conjugate margins with highs and domes observed in this domain is consequently poorly understood.”

15

L45-46: Add that you focus on the northern Iberia-Newfoundland conjugate margins, because the timing of deformation is diachronous along strike (Mohn et al., 2015; Nirrengarten et al., 2018)

We have modified the text to highlight this aspect.

“ We focus on the northernmost part of the Iberia-Newfoundland conjugate rifted margin system (Fig. 1). High-resolution conjugate seismic reflection and refraction profiles on the Flemish Cap (e.g., SCREECH-1) and Galicia conjugate margins (e.g., ISE-1/WE-1) (Fig. 1) , and sampling of peridotite rocks reveal the following key-characteristics of the transitional domain.”

We also add a sentence about the along strike rift propagation-related diachronicity of the deformation in the section “Model implications” and cite Mohn et al. (2015) and Nirrengarten et al. (2018).

16

L48-49: highly serpentinized... melting: add that it is based on seismic velocity interpretation

We have modified the text to clarify the origin of the features described here. It is well established from ODP and rock sampling that the top of the peridotite ridge is serpentinized mantle. Recent seismic refraction data confirm that most of the highs have velocities that suggest that these similarly represent serpentinized peridotite ridges (Davy et al., 2017) (See also comment 15).

17

L47-58: Could you add the different described domains on the Fig 1

We now refer specifically to ridge name and position in our description and we have added the different domains on figure 1 (see also comments n°1 and n°13).

18

L53: with elevations 500-1600 m above the seafloor: Is it an average seafloor or the deepest seafloor or the variation of topography between the base and the top of the structure?

We now modify this sentence to clarify the definition.

“...with amplitudes of 500-1600 m between the base and the top of the structures (ridges B, C, and D1 on Fig. 1a and B, C, D on Fig 1c).”

19

L54-58: Is the distal exhumed mantle domain formed by seafloor spreading? Where is the clear oceanic crust on the sections?

We hereby reproduce reply to comment n°3. The magmatic oceanic crust along the Flemish Cap profile starting between domes E1 and E2 shows typical characteristics of 5-6 thick oceanic crust and has been interpreted as the start of a magmatic spreading center (Funck et al., 2003; Hopper et al., 2004). The corresponding distal domain on the Galicia conjugate margin shows 2-3 domes (D2, E and F on figure 3c) and evidence for episodic magmatic addition. We interpret this distal domain as resulting from slow mature mid-ocean ridge spreading characterized by along strike variations in magmatic output and episodic formation of detachments faults (e.g. Escartin et al., 2008; Howell et al., 2019). This is the reason why we prefer to characterize this distal domain as the start of a mature MOR system.

We now provide more detailed information in the description. We note that we now provide a better definition of the oceanic domain at the start of the introduction, i.e. comment n°11.

20

L59; new analysis and interpretation: The interpretation of multiple detachment was already made no?

We agree. The new sentence is:

“We use 2-D thermo-mechanical geodynamic modeling of continental rifting and mantle exhumation and integrate this with a new analysis and kinematic reconstruction of the transitional domain from continental to oceanic crust on the magma-poor Galicia-Flemish Cap conjugate margins”

21

L79-80: Could you add the duration of each phases extract from the model or refer to Fig. S5

Yes. We now include a reference to each panel of figure S7 that corresponds to each phase and we also include the duration of each phase.

22

L84-85: Could you estimate the thickness of the frictional-plastic part of the model? And hence tell the rooting depth of the normal faults. Can it be compared to the depth approximated in other studies (Gillard et al., 2019)?

This is done in figure S8 and new figure 3. We have included this information in the description of the modelling results.

“ The normal faults form in sequence at high angle within the frictional-plastic part of the exhuming mantle on top of the thermal necking zone, which is characterized by a brittle thickness of about 8 km in agreement with estimates for the Flemish Cap-Galicia conjugates²¹ (Fig. 3 and supplementary Fig. S8).”

We have included a full set of models exploring the sensitivity to extension rate and strain weakening in the supplement, created a new figure 3 and moved panel c from figure S6 in the main text as new figure 7. We now show how brittle thickness varies with extension rate (New figure 3).

23

L126: from proximal to distal margin: I think that it is from the proximal to the distal transition zone no? Could you show the limit between the two domain on Fig.3

Indeed, the text has been modified accordingly.

“ Basement morphology of the transitional domain on the conjugate Galicia-Flemish Cap margins is similarly interpreted allowing to link these basement features on both conjugates in order of formation from the proximal to distal part of the transitional domain (Fig. 4b,c). While episodic magmatic activity is not included in the forward model it reproduces and explains the first order

features of this transition zone following crustal break-up and before establishment of a stable mid oceanic spreading center.”

Each basement high can be interpreted as a footwall scarp that has a corresponding fault root zone on the conjugate margin during symmetric exhumation characteristic of model behavior. This is represented on figure 4 with a number and a different color for each couple. We have added labels for proximal/distal domains on figure 4.

24

L130-139: In this three sentences you assume that the modelling is perfectly adapted to the Iberia-Newfoundland case and that the age of closure at 600° is more robust than the Ar-Ar dating of syn-tectonic plagioclase. I wonder if the model is by itself sufficient to state this.

The models are not perfectly adapted to the Iberia-Newfoundland system. However, the rate of mantle upwelling and the cooling rate during mantle exhumation are directly controlled by the extension rate. As shown by the models when extending at 1.5 cm/yr, a rate characteristic for the Iberia-Newfoundland system, a sample that passes through the 600 °C closing temperature reaches the surface in less than 1 Myr. We believe that this is a robust result.

Regarding the various closure ages from Ar/Ar on plagioclase, U/Pb on zircons, and Ar/Ar on amphibole, it is well known that diffusion of Ar in plagioclase is associated with large uncertainty and is difficult to constrain (Cassata et al., 2013). We also note that younger ages from Ar/Ar on plagioclase could be related to late fluid-induced recrystallization that causes Ar loss (Villa et al., 2006; Jagoutz et al., 2007). The other two methods used on the samples (U/Pb on zircons and Ar/Ar on amphibole) are more robust. We note that the three ages, one from U/Pb on zircon and two from Ar/Ar on amphibole, give a similar age of 122 Myr. The age Ar/Ar on plagioclase of 117 Myr would indicate a very low extension velocity of about 0.5 cm/yr that is not consistent with inferred rates (Brune et al., 2016). We believe that this apparent contradiction results from uncertainties with

Ar/Ar on plagioclase dating.

We now show the sensitivity of cooling rate with extension rate and discuss this aspect (New figure 3).

25

L139-142: The regional...breakup: Could the regional unconformity be linked to the initiation of steady state seafloor spreading? I do not understand well why this sentence is here.

We agree that there is no need to discuss the regional unconformity at this stage in the manuscript. We have removed this sentence. We instead mention that the M0 magnetic anomaly, which is dated at 121.4 Ma with a duration of 0.4 Myr, is close to crustal breakup time and that other older M1, M2, M3 magnetic anomalies pre-date crustal breakup implying that these anomalies should not be visible on the seafloor at this latitude and therefore cannot be used to constrain kinematic reconstruction.

26

L149: is recognized at 60 km: What are the criterion to determine well defined oceanic crust on Screech 1.

Hopper et al. (2004) and Funck et al. (2003) proposed based on seismic refraction velocities that there is a 5-6 km thick oceanic crust in the distal domain of the seismic profile across the Flemish Cap distal transitional domain (km 300, ~ ridge E1 on figure 1c) indicating the start of seafloor spreading (see also reply to comment n°19).

27

L150-151: with an age ..between 121-83 Ma: You should also assume that the line is oriented parallel to the flowline.

The reviewer is right. We have modified the sentence accordingly.

28

L152-153: What are the characteristics indicating the presence of magmatic additions?

We now provide a more explicit description in the text and in the caption of figure 4.

29

L156-157: The J-anomaly is a magnetic anomaly which is poly-genic and poly-phased (Nirrengarten et al., 2017) on the northern part of the conjugate margin this anomaly is less pronounced then farther south and the position of the first steady state oceanic crust is different between different authors (Dean et al., 2015; Gillard, Manatschal, et al., 2016; Nirrengarten et al., 2017; Sutra et al., 2013). At one point in the document it would be nice to see your criterion to determine steady state oceanic crust.

We have improved the definition provided for “mature steady-state spreading center” in the introduction (see reply to comment n°11):

“The transition from magma-poor spreading and mantle exhumation to establishment of a stable mature mid oceanic spreading center, defined by standard oceanic crustal thickness in the range 4-8 km and locally exhumed mantle domes for slow and ultra-slow spreading rates, appears to be gradual with progressive increase of magmatic addition towards the distal domain ”

See reply to comments n°3, n°26 and n°28 for a description of the criteria associated with a mature MOR steady-state spreading system.

We acknowledge work from Nirrengarten et al., (2017) about the J-anomaly:

“The J-anomaly appears spatially correlated with the earliest start of mid-ocean ridge spreading which suggests that these regional events are associated closely in time⁴⁶. However, we note that this correlation is not a proof of causality since the J-anomaly is a magnetic anomaly that might be polygenic and is not continuous along the distal transitional domain of the Newfoundland-Iberia system⁴⁸ (Fig. 1e).”

30

L 180 non volcanic: prior it was magma poor choose one terminology.

See also comment n°7. We agree and we now use “magma-poor” everywhere in the manuscript.

31

L186-188: The structural interpretation has already been published (Dean et al., 2015; Gillard, Manatschal, et al., 2016), the novelty of the paper is rather the determination of physical processes permitting to reproduce the interpreted architecture (increasing thermal conductivity of mantle rocks with decreasing temperature, and separation of viscous / frictional plastic strain).

We have modified the discussion to better acknowledge previous work and show the novelty of this paper.

32

L188-190: already said before

We have modified the discussion and this sentence.

33

Figure1: -The J anomaly is not punctual et rather occupies a wider zone (Whitmarsh & Miles, 1995)

- On the figure the transitional domain is called Zone of Exhumed Mantle but on the legend panel it is written may be exhumed mantle, oceanic crust, continental crust. I suggest writting exhumed mantle with potential magmatic addition or remnants of continental crust

We have modified the figure accordingly.

34

- Fig 1E Colors from the legend panel do not match the one on the map. ODP-DSDP sites instead of IODP moreover they are not plotted on the map.

We have updated the figure.

35

Figure 2b: In the presented model at the active spreading system, the top basement is around 5km below seafloor, isn't it to deep? Could you comment on that

This is a very good comment. The density structure of this model is not calibrated to fit the relative elevation between continents and mid-ocean ridge. This is actually the scope of a paper that is in revision. We have added a comment about this in the figure caption. However, this does not affect the structure of the exhumed mantle domain.

36

-L251 pre-rift sediments (purple)

Done.

37

Figure 3: How Figure 3B has been drawn? What are the reasons to add volcanism?

The magmatic oceanic crust along the Flemish Cap profile starting between domes E1 and E2 shows typical characteristics of 5-6 thick oceanic crust and has been interpreted as the start of a magmatic spreading center (Funck et al., 2003; Hopper et al., 2004). West of dome E2, Hopper et al. (2004) argue for the presence of a thin continuous 2 km oceanic crust. The domain between highs C and E1 exhibits a rough high frequency top basement topography with small amplitude that we interpret as small offset high angle faults associated with volcanism (Figure 3). We find similar characteristic top basement morphology on Galicia side that we interpret as magmatic addition.

This information is now better explained in the caption and in the text (see also reply to comment n°28).

38

- Detachment 6 do not have the same color on a and b sub-figures.
- I suggest to invert sub-figure b and c to have the most interpretative section at the bottom.
- Light green exhumed mantle could you rather show the serpentinized mantle because in a way all the mantle is exhumed from deeper level.
- Add the color code on the figure

We have updated the figure and added legends to describe the different colors. However, the green mantle does not represent serpentinized mantle but a-magmatic non melting mantle.

39

Fig S1: I think that the observed top basement (blue curve) should be corrected by flexural back-stripping. Indeed the current topography has encompassed sediments loading and thermal

subsidence, in order to compare the model basement topography and the picked top basement you should perform a back-stripping (Nirrengarten et al., 2016).

We have computed the regional flexural isostatic response to the load provided by the post-rift sediments for an elastic thickness of 35 km, which is reasonable for oceanic lithosphere of this age. The differential elevation owing to flexural loading is in the order of 100 m (Figures S2 and S3). We have mentioned this in the text (see also reply to comment n°28).

References

- Dean, S. L., Sawyer, D. S., & Morgan, J. K. (2015). Galicia Bank ocean-continent transition zone: New seismic reflection constraints. *Earth and Planetary Science Letters*, 413, 197–207. <https://doi.org/10.1016/j.epsl.2014.12.045>
- Gillard, M., Manatschal, G., & Autin, J. (2016). How can asymmetric detachment faults generate symmetric Ocean Continent Transitions? *Terra Nova*, 28(1), 27–34. <https://doi.org/10.1111/ter.12183>
- Gillard, M., Autin, J., & Manatschal, G. (2016). Fault systems at hyper-extended rifted margins and embryonic oceanic crust: Structural style, evolution and relation to magma. *Marine and Petroleum Geology*, 76, 51–67. <https://doi.org/10.1016/j.marpetgeo.2016.05.013>
- Gillard, M., Tugend, J., Müntener, O., Manatschal, G., Karner, G. D., Autin, J., ... Ulrich, M. (2019). The role of serpentinization and magmatism in the formation of decoupling interfaces at magma-poor rifted margins. *Earth-Science Reviews*, 196(June), 102882. <https://doi.org/10.1016/j.earscirev.2019.102882>
- Mohn, G., Karner, G. D., Manatschal, G., & Johnson, C. A. (2015). Structural and stratigraphic evolution of the Iberia-Newfoundland hyper-extended rifted margin: a quantitative modelling approach. Geological Society, London, Special Publications, 15, 13810. <https://doi.org/10.1144/SP413.9>

Nirrengarten, M., Manatschal, G., Yuan, X. P., Kuszniir, N. J., & Maillot, B. (2016). Application of the critical Coulomb wedge theory to hyper-extended, magma-poor rifted margins. *Earth and Planetary Science Letters*, 442. <https://doi.org/10.1016/j.epsl.2016.03.004>

Nirrengarten, M., Manatschal, G., Tugend, J., Kuszniir, N. J., & Sauter, D. (2017). Nature and origin of the J-magnetic anomaly offshore Iberia–Newfoundland: implications for plate reconstructions. *Terra Nova*, 29(1). <https://doi.org/10.1111/ter.12240>

Nirrengarten, M., Manatschal, G., Tugend, J., Kuszniir, N., & Sauter, D. (2018). Kinematic Evolution of the Southern North Atlantic: Implications for the Formation of Hyperextended Rift Systems. *Tectonics*, 37(1), 89–118. <https://doi.org/10.1002/2017TC004495>

Reston, T. J., & McDermott, K. G. (2011). Successive detachment faults and mantle unroofing at magma-poor rifted margins. *Geology*, 39(11), 1071–1074. <https://doi.org/10.1130/G32428.1>

Sutra, E., Manatschal, G., Mohn, G., & Unternehr, P. (2013). Quantification and restoration of extensional deformation along the Western Iberia and Newfoundland rifted margins. *Geochemistry, Geophysics, Geosystems*, 14(8), 2575–2597. <https://doi.org/10.1002/ggge.20135>

Whitmarsh, R. B., & Miles, P. R. (1995). Models of the Development of the West Iberia Rifted Continental-Margin at 40-Degrees-30n Deduced from Surface and Deep-Tow Magnetic-Anomalies. *Journal of Geophysical Research-Solid Earth*, 100(B3), 3789–3806. <https://doi.org/Doi 10.1029/94jb02877>

Reviewer #2 (Remarks to the Author): Gaël Lymer

Review of the manuscript NCOMMS-21-34634, Mantle exhumation at magma-poor rifted margins: a competition between frictional shear zones and thermally weakened necking domains, by Thomas Theunissen and Ritske S. Huisman

The MS submitted by T. Theunissen and R. Huisman investigates the mechanisms and structures (mantle ridges, detachments) that develop during mantle exhumation in the transition zone from continental to oceanic crust of distal magma-poor rifted margins. The authors present the results of forward geodynamic numerical models and provide explanations for the geometry and origin of exhumed mantle peridotite ridges. The authors propose that during extension, strength competition between weak frictional-plastic shear zones and thermally weakened necking domain develops, favouring the formation of multiple out-of-sequence detachments with recurring dip reversal, permitting emplacement of mantle ridges as a result of cross cut between successive detachments. The authors then use the first order characteristics of their numerical results to guide kinematic reconstruction of the transitional domain offshore the system of Newfoundland-Galicia conjugated margins pair, and to propose a model of emplacement of peridotite ridges and magma, also providing time constraints on the last stages of breakup and first oceanic crust emplacement at these margins. Specifically, the model highlights that remnants of the crosscutting detachments are now located in the exhumed mantle domain of both sides of the conjugated margins pair. The model also supports an Albian-Aptian initiation of a mature magmatic spreading centre. The authors finally propose that their model also provides explanations for emplacement of similar structures at the South-West Indian Ridge and the Cayman spreading centre, suggesting that their numerical model can be applied to other magma-poor margins and magma poor spreading centres, that strongly broadens the significance of the study.

We really appreciate the very detailed comments from Dr Gaël Lymer. We sometimes refer to a previous or a next answer to avoid repetition.

1

I have a good knowledge of the structures and nature of the rock materials at the Galicia-Newfoundland conjugated margins from geophysical and sampling data, but I am not a numerical modeller and my knowledge of this approach is based on work published by other researchers. I find, the MS very well written and of very good quality with clear explanations of the context and appropriately citing references. The work is well described and should be possible to reproduce by others. The presented data and figures, both in main text and in supplementary, are suitable to introduce the first order tectonic structures observed at the studied margins and to support the analytical approach and discussion of the MS. The numerical model and associated parameters look robust and valid to me: the proposed model, in my opinion, convincingly reproduce the first order structures of the mantle domain at the Galicia-Newfoundland margins, as presented in fig 1. Predicted age of exhumation of the first PR on the Galicia side is right in the range of estimated age resulting from IODP sampling, and the predicted age of 113.8 ± 1.6 Ma for the start of a mature magmatic spreading center is consistent with the generally agreed Albian-Aptian age of breakup, further supporting the proposed numerical model. The proposed explanations for the mechanisms and kinematics of emplacement of the structures observed at the distal domain of the Galicia-Newfoundland margins appears robust and consistent with existing data and knowledge at this margin, as well as with structural model proposed for the emplacement of crosscutting detachments at oceanic spreading centre (e.g. Reston, 2018), thus strongly supporting the conclusions of the MS.

We appreciate the positive assessment of our results.

2

I have, however, two main questions regarding 1) the effect of uniform (1.5 cm/yr) extension velocity used in the models on the formed structures, and 2) on the change of faulting regime from in-sequence faulting during crustal hyper-extension to out-of-sequence during mantle exhumation, please see details below (main questions highlighted in bold). I also suggest below minor editions of the main text and figures and ask some more minor questions.

I hope my feedback will be useful and will help reviewing and publishing this MS.

Wishing to the authors good luck in the next stages of editions of the MS.

Best regards,

Gaël Lymer

We directly answer to these two major questions in comments n°9 and n°10 respectively.

3

Main text:

L. 53: “asymmetric ridges with elevations 500-1600 m above the seafloor”

This suggests to me that all ridges outcrop above the seafloor, but not all the ridges actually reach the seafloor, and only 2 on Fig. 1. Elevations laterally change a lot along the ridges in 3D. Maybe precise the sentence?

We now modify this sentence to clarify the definition.

“...with amplitudes of 500-1600 m between the base and the top of the structures (ridges B, C, and D1 on Fig. 1a and B, C, D on Fig 1c).”

4

L.55: “and (5) several basement highs between 1000-2300 m elevated above the seafloor”

It is not clear to me where this is, is it in the oceanic domain? Maybe precise which basement you refer to and/or point them on figure 1?

We have updated the sentence.

“(5) A distal transitional domain on the Galicia side with evidence of episodic magmatic activity and discontinuous stretches of thin oceanic crust with smoother basement topography, and (6) several basement highs with amplitudes of 1000-2300 m (Galicia: ridges D2, E and F, e.g., Fig. 1a).”

5

L.84-97: I think it would help the reader if the description of the model provided in this section of the main text was supported by visual markers on the figures (Fig. 2, Fig. S5) pointing to the features described in the text.

We have included the duration of each phase with a reference to figure S7 and included reference to Figure 4a where detachment faults are numbered and to the animation of model M1 where the sequence of events described in the text is well visible.

6

L.151: “On the Galician margin ridge 3a is at approximately 60 km distance from the most distal location of continental crust”

I cannot find ridge 3a. Is it ridge D1 on fig.2? Or ridge 5 on Fig. 3? Please check.

Thank you to have noticed this mistake.

“On the Galician margin the symmetry of the system implies that ridge D1, which is at approximately 60 km distance from the most distal location of continental crust, represents the start of a mature magmatic spreading center on the conjugate.”

7

L.153: “On average 700-800 shallower basement can be explained with the isostatic consequence of 6 km thick oceanic crust (Fig. S1).”

Can volume-increase magmatic additions also contribute to shallow this domain?

We are not fully sure we understand. However, the volume or thickness of lower density oceanic crust changes the isostatic balance. A higher thickness of oceanic crust results in a shallower bathymetry.

8

L. 207: “Our models provide therefore a framework for understanding formation of both the exhumed mantle along non-volcanic rifted margins and for the a-magmatic segments of ultra-slow mid-ocean ridge systems.”

Reston 2018 also provided explanations for mid-ocean ridge systems in a sensibly similar way, although based on structural analysis rather than numerical simulation. Maybe also refer to his work at this place? (currently only cited line 37 if I am correct).

Yes we agree that Reston (2018) proposed systematic flip-flop mode of faulting at ultra-slow spreading systems. His conceptual models also includes the effect of melt extraction on the mode of faulting and strain migration, in contrast to our models that do not include the effect of melting. We have cited Reston (2018) in the discussion and have modified the last part of the discussion.

9

Figures:

Fig. 2 (Also Figure S5, Table S2 and supplement movies S1 & S2):

Main question 1: In the presented model, the extension occurred at uniform velocity of 1.5 cm/yr.

Other studies have predicted an acceleration of extension before continental break-up, with a transition from slow to fast extension during hyper extension, development of detachment and mantle exhumation (Brune et al. 2016, Abrupt plate accelerations shape rifted continental margins, Nature volume 536, pages 201–204).

As rift evolution appears strain-rate-dependent, I wonder what would be the implication in your models of a uniform vs accelerating velocity of extension on the development of structures in the distal margins? Have you made models with varying velocities?

Brune et al. (2016) inferred an acceleration of the rate of extension from 0.5 to 1.5 cm/yr at about 140 Ma with subsequent constant extension rate until about 110 Ma. This change in rifting velocity occurred therefore well before crustal breakup of the Iberia-Newfoundland system. If anything, this may have affected the progressive localization of deformation but is not relevant for the exhumed mantle domain. We are using a uniform extension rate but we now include a full set of sensitivity tests in the supplement. These sensitivity tests show the effect of varying spreading rates between 1.5 and 0.7 cm/yr on the characteristics of the exhumed mantle domain including brittle thickness and sea floor morphology. (New figures 3 and S9-15).

10

Main question 2: The structures with fault-bounded peridotite ridges obtained in the final stage of the forward model strongly resemble the wide zone of exhumed mantle observed from seismic data at the Galicia margin. It is also consistent with proposed models for tectonic process of accretion of oceanic crust at mid-oceanic ridges (e.g. Reston, 2018).

Although the authors clearly specify that the focus of this paper is the exhumed mantle domain, differences exist between the presented model and the observed structures of the hyper-extended

domain of the Galicia margin at the edge of the continental crust. There, at least in the area of location of line WE-1 shown in Fig. 1, the block bounding faults are systematically dipping toward the ocean over >30 km above the S detachment (e.g. Lymer et al. 2019). This does not seem to be simulated in the presented model.

The authors refer to previous publications for the development of the necking domain and distal margin (L. 79), but I think the paper should be self-consistent and I wonder if the change of faulting regime from synthetic faults in the hyper-extended domain above S detachment, to out-of-sequence faulting in the exhumed mantle domain, resulting in flip-flop detachments, should be discussed or at least mentioned in this paper? Also, could this change be related to the aforementioned variations of velocity of extension?

The Iberia-Newfoundland system is characterized by a long complex rift evolution with multiple phases of rifting in the Triassic, Jurassic, and Cretaceous. The Northern Galicia margin including the Galicia Interior basin experienced extension at least in the earlier Jurassic phases resulting in a complex rift and crustal structure at the start of the final phase of rifting with crust thinned to about 15 km as observed at present day at Galicia Bank. This pre-existing crustal template controlled the architecture of the distal margin. It is beyond the scope of this paper to resolve these additional complexities. We note that mantle serpentinization resulting from fluids reaching the mantle during the last phase of rifting may also play a role in forming a detachment at the base of the crust.

However, while these are certainly interesting aspects, reproducing the most distal part of the crustal domain is not the focus of this manuscript. We now mention this in the model description as a limitation of the modeling approach.

“We note that we did not aim to reproduce the detailed style of faulting during the formation of the distal continental margin with ocean-ward dipping faults rooting in the S detachment (Lymer et al., 2019).”

Fig. 3 & Fig. 5: Please highlight Galicia side and Newfoundland side.

A confusing point to me here is that the S detachment on the Galicia side is not indicated. Instead on figs 3 & 5, detachment 1 (red (unofficial colour for S in different publications)) appears to underline continental crust on the Newfoundland side (Fig. 3b), as does the S on the Galicia side. On figs 3&5, detachment 1 under continental crust is shown oceanward of the COB, while no continental crust is shown oceanward of the COB on Fig1C,D (although the key clearly says the transitional domain can have various nature). Figs 3 (km 200) & 5 currently display a black fault on the Galicia side, which is not mentioned in the text.

In summary: Detachment 1 currently looks to me like the S detachment on the Galicia side, making figures 3&5 looking like if they were in the wrong sense (i.e. Galicia on the left).

I think these should be clarified; some ideas: highlight Galicia/Newfoundland; change the colour of the material above detachment 1 on the NWFDL side from crust to mantle or undefined? Show the S on the Galicia side? define the black fault?

We have modified figures 4 and 6 to highlight Galicia/Flemish Cap sides. We now indicate the S-reflector in figure 4. We note that the highly thinned crust in the distal Flemish Cap transect is interpreted in agreement with Sutra and Manatschal (2013). These points are clarified in the captions and in the discussion (see also reply to comment n°10).

12

Fig. S1: Can you precise how you obtained the basement plot (blue curve), is it interpretation from figure 2 or from a reference?

The blue curve is the elevation of the top basement as represented in Figures 1a, 1c and 2c. This is now clarified in the caption of Figure S1.

13

Fig. S3: The inherited weak domain instead of a single seed, allowing more freedom for fault development is a great idea! Such “more freedom” parameter was missing from other publications I have read on numerical models.

This approach has been used in previous publications by some authors (Huismans et Beaumont, 2007; Theunissen and Huismans, 2019; Brune et al., 2014; Duclaux et al., 2019).

14

Tables S1 & S2: Except my point on uniform velocity of extension (see comment for fig. 2), the bounding conditions of table 2 and mechanical and thermal conditions of table 1 look good to me, although I am no specialist on this matter.

We have now included the full set of sensitivity tests including effect of varying extension rate and varying strain weakening parameters to strengthen our conclusions.

Reviewer #3 (Remarks to the Author): Laetitia Le Pourhiet

Review to Mantle exhumation at magma-poor rifted margins: a competition between frictional shear zones and thermally activated necking domains by T. Theunissen and R.S. Huisman

I have read the paper and supplementary material carefully and I must say that it is difficult to evaluate the modelling part of the manuscript, as I was asked to, because there is only one finely tuned model presented and it is impossible to evaluate how the spacing and timing of the flip-flop depends on softening, rate of spreading etc.. As the timing is critical for the kinematic reconstruction at the end, it seems to me that the authors should demonstrate they have a robust model and they need to display how the change in the parameters affects the solution.

We really appreciate the very detailed and constructive comments. We sometimes refer to a previous or a next answer to avoid repetition. We now include a comprehensive set of supplementary models that show the sensitivity to parameter variations.

1

In the current form, I really don't see what "your model add to current understanding of non volcanic margin and ultra slow mid-oceanic ridge system" as you conclude. I agree that the model (you could call M instead of M1 as there is only one model) produces flip-flops... but the flip flop conceptual model is established for a long time, and the parallel between magma poor margins and slow spreading ridge is in my opinion textbook material (master level) today but I might be biased by my proximity to M. Cannat and S. Leroy. The big questions are how to produce rolling hinge down to 16 km depth not 7 or 10 km (e.g. Bickert et al. 2020), and how it relates to melt. Any simulations with softening of the mantle material and no melt can produce flip-flop (see per exemple Jourdon et al 2019 : 5 lines in a paper "flip-flop are produced consistent to observations at

Galicia margin and SWIR”) . Taras Gerya family models generally don't produce the flip flop because they are all coupled with melt as he really intend to model oceanic crust.

We agree that the flip-flop conceptual model and observations showing the parallel between magma-poor margins with slow and ultra-slow spreading ridge are established for a long time (e.g. Reston and McDermott, 2011; Cannat et al., 2009; Sauter et al., 2013; Gillard et al., 2016a, 2016b; Reston, 2018). However, as also clearly stated by reviewer 1 and 2 no geodynamic model up to now was able to reproduce these observations. The models presented here are the first to self-consistently produce detachment faulting with alternating directions explaining characteristic features observed along the Iberia-Newfoundland magma-poor conjugate margins including wavelength, topography, and brittle layer thickness. The models provide also new understanding how footwall scarps and fault root zones are typically separated during spreading allowing linking the conjugate margins. We use this to establish a new reconstruction for the exhumed mantle domain along the conjugate Galicia-Flemish Cap margins. The models are furthermore used to infer mantle cooling rate during the final stages of rifting allowing to constrain the time of crustal breakup and the timing of establishment of a mature magmatic mid-ocean ridge spreading center.

We do think that it is well established that strain weakening is required to form single large offset rolling-hinge fault zones (Lavie et al., 1999, 2000). Buck et al. (2005) showed in addition that slow spreading mid-ocean ridge systems are dominated by tectonic spreading and that extension is accommodated by large offset normal faults when magmatic spreading accommodates less than 50% of the extension ($M < 0.5$). Similar studies investigate the mode of faulting with varying M or parameters controlling the prescribed thermal structure (e.g. Behn and Ito 2008; Ito and Behn, 2008; Tucholke et al., 2008; Liu and Buck, 2018; Howell et al., 2019). There is limited fault interaction in these models when $M > 0.3$ with predominantly in-sequence fault migration. For the a-magmatic case ($M = 0$), Tucholke et al. (2008) shows irregular out-of-sequence deformation of short-offset faults at intermediate spreading rate (50 mm/yr) using a prescribed brittle thickness of 5 km. Bickert

et al. (2020) included in addition to cohesion weakening other weakening processes (serpentinization and viscous weakening at high stress and high strain rate) using the same modeling approach resulting in flip-flop mode of faulting for ultra-slow spreading rate (14 mm/yr) and a very thick prescribed brittle layer at the axis (20-25 km). Lastly Pütke and Gerya (2014) and Gülcher et al. (2019) have used a more self-consistent modelling approach of slow mid-ocean ridge spreading systems (1.5-2.5 cm/yr) that includes melting, formation of a magma-chamber and oceanic crust formation resulting in models that exhibit detachment faulting and out-of-sequence migration but these studies are not directly relevant for magma-poor systems. None of these studies explained the formation of multiple large offset normal faults with alternating directions that reproduce basement topography as observed in these magma-poor systems which is the focus of our work. We are aware that Jourdon et al. (2019) inferred based on 2-D thermo-mechanical modelling mantle exhumation through detachment faulting and strain migration in a flip-flop mode. However, these models do not clearly reproduce detachment faulting and neither provide an explanation for their formation while modelling a narrow exhumed mantle domain at ultra-slow spreading rates (4-6 mm/yr).

Model 1 and supplementary models included in the revised version of our manuscript now show how this flip-flop large offset faulting characteristic for magma-poor margins and slow to ultra-slow spreading depends and changes with varying the degree of strain weakening and spreading rate.

2

When I accepted to review the paper and I saw the authors which I respect, I was really expecting a complete parametric study to be present in the supplementary material so that the flip-flop behavior in model without melt could be considered as a nailed thing that we do not need to model anymore. The only thing I found was “the main changes between model M1 as compare to other published models are the conductivity of the mantle and some fine tuning of the softening parameters”. How these two parameters affect the prediction of the simulations ? Well this is key if you want to say

that modelling has brought something to our understanding of Galicia margin because you need to make sure your timing between flipflop are robust to build your kinematic model and attribute what is missing to the melt that is absent of your simulations.

We have now included a full set of models exploring sensitivity to strain weakening, full extension rate, and thermal conductivity in the supplement. We have modified the introduction and the discussion to better acknowledge previous work and use the sensitivity analysis to strengthen our conclusions. We now discuss how frictional softening parameters, spreading rate and thermal conductivity affect model behavior. The effect of melt on deformation is also discussed in the discussion.

3

Also a parametric study would have been interesting in order to better assess if the model obtained explains how to form detachment that are brittle down to 14km depth like observed at ultra-slow spreading ridge or if there is a need for new ductile laws like proposed by the modelling study of Bickert et al. 2020. In other words, if the same simulation is run with a slower rate like 7mm/yr of extension does it produce deeper detachment? (I would argue yes, maybe because in Jourdon et al. they seem to be deeper and the extension rate is slower) but still this is not in this paper while there is a figure that show how they deepens with rate in nature in the supplementary material.

We now include supplementary models that show the combined effect of varying spreading rate and fault strength on the mode of faulting, seafloor topography, brittle thickness and cooling rate during magma-poor mantle exhumation to the seafloor. Varying extension rate between 0.7 and 1.5 cm/yr results in brittle layer thickness varying between 17 and 7-8 km (New figure 3). Indeed, deeper detachments/large offset normal faults are formed for low extension rates as well. Main characteristics that change with extension rate and brittle thickness are magnitude of footwall topography and characteristic wavelength and cooling rate during exhumation. These models show

that there is no need for a new ductile law as proposed by Bickert et al. (2020). We have included a new figure demonstrating this sensitivity in the main text (New figure 3) and supplementary figures showing snapshots for each model (Figures S9-15). We have included new paragraphs in the discussion that explain how frictional softening parameters, spreading rate and thermal conductivity affect model behavior.

4

To me it seems this paper has been redirected straight from a shorter format nature family journal because the authors cannot really make their point in a short format. So I would recommend the paper for major revision to give the author the time to actually move some of the modelling material like very good figure S6 to be in the main text but also to actually push the parametric study in the suppl. Material and partly in the main text to strengthen their new timing arguments about the Galicia margin. I am sure that the authors, who are serious scientists, did perform this parametric study for them-selves so it should not be a lot of extra work and I think it is important that the reader and the reviewer can assess how the best fitting model is sensitive to variation of new parameters and rate of extension. Below I also list some minor comments.

With best regards

Laetitia Le Pourhiet

We are happy with the very constructive comments. We did perform a large parametric study but decided for brevity to leave this out in the first submission. We have now included a parametric set of models in the supplement and discuss sensitivity to the main controlling parameters in the main text.

===== minor comments =====

I don't understand why the crustal material softens more than the mantle... I would have expected the opposite as the mantle goes through serpentinisation while the continental crust does not produce phyllosilicates as weak as talc and serpentines. How does this choice affect your result ?

We understand that this is confusing. In the models presented here we have focused on calibrating the degree of strain weakening in the mantle to reproduce the observed structures in the exhumed mantle domain along the Iberia-Newfoundland conjugate margins. In order to reproduce observed symmetry of this domain between the two conjugate margins, the wavelength, and the topography of peridotite ridges we have needed to limit the degree of strain weakening in the mantle materials. We do acknowledge that the plastic strain weakening in our models is parametric representing a range of complex physical processes including (but not limited to) mineral transformations, effects of fluid overpressure, and possibly viscous weakening or hardening around the brittle-ductile transition. However, we are confident that the parametric calibration of weakening for the mantle domain is robust as it allows reproducing observed characteristics.

We do note that the weakening parameters of continental crust are based on previous modelling work and may benefit from further calibration. Adopting lower weakening for the crust would result in slightly different fault network geometry but not change the main insights gained here on mantle exhumation.

We also note the study of Hansen et al. (2019) that demonstrates significant viscous strain hardening following grain size reduction for olivine at conditions characteristic for the brittle-ductile transition. While we have not included this additional complexity, it may provide an explanation for reduced effective strain weakening in mantle materials. We also note previous

modelling works that used lower degrees of plastic (brittle) strain weakening in mantle materials (Choi and Lavier, 2013; Püthe and Gerya, 2014, Gulcher et al., 2019).

We added this aspect in the “Material and Methods – Fault strength” section.

6

How the lack of melt and its feedback on deformation in the formulation of the model affects the result. I can see from the snapshots that the isotherm 1300°C is at 30-35 km depth from 12-16 Ma on. In these conditions the mantle would probably melt (and you do propose it does on the cartoon of Fig 6) so it would be good to actually at least post process the melt produced by the model and discuss how its presence would affect the dynamic of the flip-flop. Moreover computing the actual timing of melt would be of great help to calibrate your simulation to the Galicia margin and strengthen your arguments about kinematic reconstruction.

A large range of observational studies demonstrates that the exhumed mantle domain along the Iberia-Newfoundland conjugate margins is largely a-magmatic (e.g., Jagoutz et al., 2007; Cannat et al., 2009; Gillard et al., 2019; Granado et al., 2021). Only in the distal part of the exhumed mantle domain there is low degree of episodic magmatic addition expressed as a thin ~2 km oceanic crust before the establishment of a mature mid-oceanic spreading system with ~6 km magmatic crust (e.g., Hopper et al., 2004). Various mechanisms have been proposed to explain the magma poor nature of the exhumed mantle domain (e.g., Tugend et al., 2018 and references therein) including (but not limited to) low mantle potential temperature (Minshull et al., 2001), ultra-slow spreading rate, and counter-flow of depleted lower lithospheric mantle (Huisman and Beaumont, 2011; Beaumont and Ings, 2012). As the degree of magmatism is very low to absent we have chosen to ignore mantle melting in our modelling approach. We believe that counter-flow of inherited depleted mantle is the most viable hypothesis to explain the absence of mantle melting as there is evidence for depleted mantle in the exhumed mantle domain from ODP/DSDP sampling and

dredging (e.g., Muntener et Manatschal, 2006; Cannat et al., 2009; Manatschal and Lavier, 2015), as there is no evidence for an anomalous low mantle temperature (Minshull et al., 2001), and as inferred extension rates for the final phase of rifting and mantle exhumation are in the order of 1.5 cm/yr full spreading rate sufficient for decompression melting (this study; Brune et al., 2016).

In our models we have chosen for simplicity not to include counter-flow of depleted lower lithospheric mantle and have opted instead to disable mantle melting. The exhumed mantle domain in our models can be interpreted as representing depleted mantle that does not produce decompression melting. We do note this now in the discussion and in the method section.

We can speculate how low degree of melting may affect model behavior based on for instance previous work by Buck et al. (2005) that shows that at low degree of magmatic spreading, tectonic spreading dominates with large offset normal faults (as in our models) and that higher degree of magmatic spreading favors formation of multiple small offset normal faults. Fully consistent two phase flow modelling including the interaction of magmatism and solid deformation would allow to understand the interaction but this is clearly beyond the scope of our study, and given the observational evidences not a critical missing factor.

We added this aspect in the “Material and Methods – Neglecting magma supply” section.

7

The model has no erosion sedimentation could you discuss how sedimentation would affect the relief observed on the margin. I expect they could grow larger with sediment infill.

Yes we agree that this is an interesting point. We are well aware that sedimentation enhances fault offset (Olive et al., 2014; Theunissen et al., 2019; De Sagazan and Olive, 2020). The Iberia-Newfoundland system is largely sediment starved during rifting and early spreading. Similarly, only

thin hemi-pelagic deposits characterize slow and ultra-slow mid-ocean ridge spreading systems. We therefore have not included deposition in these models. However, we expect that adding deposition in these models would enhance offset of normal faults and modify fault network geometry.

8

At line 137, you suggest that your model is correct and that Argon dating is be wrong. There are many ways for your model not to be correct, including thermal blanketing by sediments which would delay cooling. I would just say that your model does not capture the slow cooling suggested by the ArAr age but that this could be due either to some approximation in your model or to the issue of diffusion in Ar Ar dating.

Iberia-Newfoundland conjugate passive margins form a sediment-starved system during the entire syn-rift. As a consequence, sediment blanketing effect is not relevant.

The models are not perfectly adapted to the Iberia-Newfoundland system. However, the rate of mantle upwelling and cooling rate during mantle exhumation are directly controlled by the extension rate. As shown by the models when extending at 1.5 cm/yr, a rate characteristic for the Iberia-Newfoundland system, a sample that passes through the 600°C closing temperature reached the surface in less than 1 Myr. We believe that this is a robust result.

Regarding the various closure ages from Ar/Ar on plagioclase, U/Pb on zircons, and Ar/Ar on amphibole, it is well known that diffusion of Ar in plagioclase is associated with large uncertainty and is difficult to constrain (Cassata et al., 2013). We also note that younger ages from Ar/Ar on plagioclase could be related to late fluid-induced recrystallization that causes Ar loss (Villa et al., 2006; Jagoutz et al., 2007). The other two methods used on the samples (U/Pb on zircons and Ar/Ar on amphibole) are more robust. We note that the three ages, one from U/Pb on zircon and two from Ar/Ar on amphibole, give a similar age of 122 Myr. The age Ar/Ar on plagioclase with a closure

temperature around 200°C of 117 Myr would indicate a very low extension velocity of about 0.5 cm/yr that is not consistent with inferred rates (this study; Brune et al., 2016). We believe that this apparent contradiction results from uncertainties with Ar/Ar on plagioclase dating.

We now show the sensitivity of cooling rate with extension rate and better discuss this aspect (New figure 3).

9

At line 139 : your interpretation of regional unconformity is not based on your modelling result since you do not simulate melt. Please write this interpretation is not based on your model in the text.

This is correct. We have removed this sentence as there is no need to discuss the regional unconformity at this stage in the manuscript. We instead mention that the M0 magnetic anomaly, which is dated at 121.4 Ma with a duration of 0.4 Myr, is close to crustal breakup time and that other older M1, M2, M3 magnetic anomalies pre-date crustal breakup implying that these anomalies should not be visible on the seafloor at this latitude and therefore cannot be used to constrain kinematic reconstruction.

10

Paragraphe l 144 to 157 : I really don't see how your model result participate to reach your conclusion... all your arguments are based on kinematics and data.

We understand that the argument we presented to infer the average spreading between the time of crustal breakup and the age of the first undisputed magmatic anomaly C34n was not completely clear. We use the age of crustal breakup based on the cooling ages established in the previous paragraph as ~121 Ma, the age of anomaly C34n given by 83 Ma, and the measured distance

between the most distal edge of continental crust and anomaly C34n. This provides an average spreading velocity of 1.6 cm/yr for the time interval [121-83] Ma. Assuming symmetric spreading, we use this velocity and the distance from the last continental block to infer an age. We have updated the text to make this clearer (See also reply to next comment n°11).

11

Paragraph l 159 to 176...

I am not sure I understand well the reasoning and if I do, I think this paragraph should come earlier in the paper and it should be make clearer to the reader.

Basically, do you assume :

- 1/ that your model without magma provides a robust timing (this should be proved by a complete parametric study that shows that the timing between to detachment faults is independent of the softening parameters and the rate of extension when you get the spacing correct) and that thanks to your model one can just count the detachment to get back to time.
- 2/ that when the detachment are not spaced like in you numerical model then the rest of extension is magmatic addition (without modelling them or at least predicting from the TM when, where and how much melt is produced)

We believe that this paragraph is in the right place in the manuscript. We have expanded the explanation and discussion of the reconstruction for the Galicia-Flemish Cap conjugate section so that the basic assumptions and reasoning are clearer.

We make two basic assumptions guided by the model behavior. (1) We assume that each basement high can be interpreted as a footwall scarp that has a corresponding fault root zone on the conjugate margin during symmetric exhumation, which is characteristic of model behavior. (2) We assume that the conjugate margin system formed symmetrically consistent with the distribution of exhumed mantle on the conjugate margins and we use the inferred average velocity (see reply to comment

n°10). With these assumptions the distance from the most distal location of continental crust equates the time/age.

The second part of the reasoning concerns the relative amount of tectonic versus magmatic spreading that is based on the characteristic offset of the normal faults in the models, which is ~15 km. Given that we interpret 7 large offset normal faults the tectonic spreading amounts to ~105 km. The total width of the exhumed mantle domain is 140 km which leaves ~35 km of magmatic spreading (i.e. 25%). We acknowledge that this is an average estimate with large uncertainty and that the magmatic spreading is more important in the distal than in the proximal exhumed mantle domain.

12

With the result you present here you cannot claim that your model also explains the ultra slow segment of the SWIR with detachment down to 16 km depth, because you do not show any model with flip flop that root at 16 km and you know as well as I know that it is very difficult to operate a rolling hinge exhumation in a layer with such a plastic thickness.

Instead of suggesting that the differences are due to the rate, please provide a model with a lower rate of extension and actually demonstrate it by including the parametric study you must have run to get the perfect model 1. Compare also your approach to the paper of Bickert et al. which does capture the 16 km depth and the spacing but need to include complex plastic behavior based on actual thin sections of sample dragged at the ridge to manage to root the decollement as deep.

As part of the parametric set of models we now show cases that are representative of ultra-slow spreading systems at rates in the order of 0.7 cm/yr with a brittle layer thickness of ~17 km (New figure 3). These models have detachments/large offset normal faults down to the base of the brittle layer and exhibit flip-flop behavior. The main features that change as noted in reply to comment 3

are the footwall topography and characteristic wavelength of the highs and the cooling rate during exhumation.

We have had a close look at the paper by Bickert et al. (2020). It is very challenging to reconstruct what weakening parameters they have used. They include (1) cohesion weakening of peridotite rocks, however, with unknown magnitude (not provided in the paper), (2) serpentinization weakening that depends on the total work (e.g., product of second invariants of stress with total strain) with a threshold of 10^8 J for temperature lower than 350°C resulting in friction angle reducing from 0.6 to 0.3 and activation energy of the dry olivine dislocation creep flow law reducing from $5.4 \cdot 10^5$ to $3.4 \cdot 10^5$ J.mol⁻¹ and (3) a dissipation dependent threshold at which they switch from a dry olivine dislocation flow law to a grain size dependent weaker diffusion flow law for temperatures between 800°C and 1000°C . As a result the shear zones in their models combine three different weakening mechanisms that results in shear localization. Given the complexity of this formulation it is very difficult to establish the effective weakening in these models, that is the strength contrast between host rock and shear zone, a quantity that could potentially be compared with our modelling results.

We do note that we find the sudden transition from a strong dislocation flow law to a weak grain size dependent flow law arbitrary. Rheological considerations and modeling suggest that viscous weakening likely plays a subordinate role during localization of large shear zones for several reasons. Most importantly, one of the main mechanisms often invoked to explain weakening of viscous shear zones, grain size reduction during dynamic recrystallization, can only play a limited role in viscous weakening as it occurs during power law creep that is insensitive to grain size, and the linear diffusion creep domain where viscous flow stress is sensitive to grain size allows for annealing and associated increase of the grain size limiting the weakening effect.

We also note that Bickert et al. impose a brittle layer thickness of 24 km, while spreading at 1.4 cm/yr full spreading rate. This combination of brittle layer thickness and spreading rate appears inconsistent with both observations that indicate a brittle layer thickness in the order of 7-8 km for

this spreading rate, and with the brittle layer thickness predicted in our models for this same spreading rate (e.g. 7 km).

REVIEWERS' COMMENTS

Reviewer #2 (Remarks to the Author):

Review 2 of the manuscript NCOMMS-21-34634A, Mantle exhumation at magma-poor rifted margins: a competition between frictional shear zones and thermally weakened necking domains, by Thomas Theunissen and Ritske S. Huisman

This is the second review of this MS investigating the development of detachments in the mantle exhumation and oceanic domains of the conjugated Flemish Cap (FCM) and Galicia (GM) magma-poor rifted margins with forward geodynamic numerical models. The authors provide explanations for the mechanisms and kinematics of emplacement of domes of exhumed mantles at these margins by the development of alternating opposite dipping detachments, and suggest that similar mechanisms can occur at other magma-poor, slow and ultra-slow extensional systems, particularly oceanic spreading centres.

I already thought that the manuscript would make a good paper at my first review, and this second version has been even more improved. As said in review 1, I am not a numerical modeller and can mainly comment on the similarities between the structures obtained by the numerical model and those shown from natural data: for me, numerical model M1 and associated kinematics presented in this manuscript are convincing. The authors have answered almost all my questions and comments from review 1, they particularly integrated new convincing figures and sensitivity analysis of model behaviour for variable spreading rates, that was one of my requests. They also addressed my other more minor comments.

I still have only one concern on a point that I raised during review 1, which is the pre-break up extension and structures. The authors clearly state that pre-break up extension is beyond the scope of this paper, so I will consider only the break-up stage of model M1 shown in Fig 6i that displays a detachment (1/1') underlying hyper-extended crust at the FCM dipping East from the FCM towards the GM, and exhuming ridge A.

The authors compare detachment 1/1' to the one interpreted by Sutra et al 2013 at the FCM, and I agree that both structures look geometrically similar, but I am more curious about their kinematics: at my knowledge Sutra et al 2013 interpreted the FCM detachment as the distal segment of the West dipping S detachment developed at the GM (see their fig 6a), and not as a newly developed East dipping detachment as supported by model M1; in other words the break-away zone of detachment 1 in this study corresponds to the root zone of the detachment in Sutra et al 2013. The FCM detachment is also interpreted as the west dipping root zone of S in Hopper et al., 2004 and Reston et al., 2007.

FCM detachment 1 in this study develops towards the east after break-up and exhumes ridge A with detachment 2, while in other papers the FCM detachment developed towards the west before the break-up and did not exhume ridge A.

I think the scenario proposed in model 1 is a possibility, but the authors should be precise in the text that the kinematics of their detachment 1 differs from the aforementioned papers, and my point should not prevent to publish this MS.

Two details: 1) maybe check again the consistency of magma-poor/a-magmatic/non-volcanic throughout the text; 2) Caption of Figure 2 refers to Figure S5 for full time evolution, it is Fig. S7.

Wishing to the authors good luck to finalize the MS.

Best regards,

Gaël Lymer

PS: My apologize for the delay in my review, I have got the Covid last week that prevented me to review this MS in time.

Reviewer #3 (Remarks to the Author):

As the authors point out, their manuscript is the first manuscript that really focus on how to make flip flop in

a numerical model of spreading. This is of a wide interest outside the modelling community as it provides new constrain on sea floor spreading in magma poor conditions. In the first shorter version of the ms, they were not providing a complete parametric study which was weakening their point. This has been corrected in this new, longer version, both in the main body of the paper and in the supplementary material. They have adress all my point and I thank them for that. These addings strengthen sufficiently their conclusions for the paper to be published with no further revisions from my modeller point of view.

Laetitia Le Pourhiet

REVIEWERS' COMMENTS

Reviewer #2 (Remarks to the Author):

Review 2 of the manuscript NCOMMS-21-34634A, Mantle exhumation at magma-poor rifted margins: a competition between frictional shear zones and thermally weakened necking domains, by Thomas Theunissen and Ritske S. Huisman

This is the second review of this MS investigating the development of detachments in the mantle exhumation and oceanic domains of the conjugated Flemish Cap (FCM) and Galicia (GM) magma-poor rifted margins with forward geodynamic numerical models. The authors provide explanations for the mechanisms and kinematics of emplacement of domes of exhumed mantles at these margins by the development of alternating opposite dipping detachments, and suggest that similar mechanisms can occur at other magma-poor, slow and ultra-slow extensional systems, particularly oceanic spreading centres.

I already thought that the manuscript would make a good paper at my first review, and this second version has been even more improved. As said in review 1, I am not a numerical modeller and can mainly comment on the similarities between the structures obtained by the numerical model and those shown from natural data: for me, numerical model M1 and associated kinematics presented in this manuscript are convincing. The authors have answered almost all my questions and comments from review 1, they particularly integrated new convincing figures and sensitivity analysis of model behaviour for variable spreading rates, that was one of my requests. They also addressed my other more minor comments.

We appreciate the very positive comments.

I still have only one concern on a point that I raised during review 1, which is the pre-break up extension and structures. The authors clearly state that pre-break up extension is beyond the scope of this paper, so I will consider only the break-up stage of model M1 shown in Fig 6i that displays a detachment (1/1') underlying hyper-extended crust at the FCM dipping East from the FCM towards the GM, and exhuming ridge A.

The authors compare detachment 1/1' to the one interpreted by Sutra et al 2013 at the FCM, and I agree that both structures look geometrically similar, but I am more curious about their kinematics: at my knowledge Sutra et al 2013 interpreted the FCM detachment as the distal segment of the West dipping S detachment developed at the GM (see their fig 6a), and not as a newly developed East dipping detachment as supported by model M1; in other words the break-away zone of detachment 1 in this study corresponds to the root zone of the detachment in Sutra et al 2013. The FCM detachment is also interpreted as the west dipping root zone of S in Hopper et al., 2004 and Reston et al., 2007.

FCM detachment 1 in this study develops towards the east after break-up and exhumes ridge A with detachment 2, while in other papers the FCM detachment developed towards the west before the break-up and did not exhume ridge A.

I think the scenario proposed in model 1 is a possibility, but the authors should precise in the text that the kinematics of their detachment 1 differs from the aforementioned papers, and my point should not prevent to publish this MS.

We agree that we should specify in the text that the geometry and the kinematic of detachment faulting at breakup time differs from previous interpretations. Sutra et al. (2013) similarly interpreted the ultra-distal blocks on the Flemish Cap margin to be continental rider blocks but they considered them to be kinematically linked to the root zone of the S detachment that can be found on the ultra-distal Galicia conjugate margin. Reston (2007), Ranero and Perez-Gussinye (2010) and Lymer et al. (2019) also suggested the same interpretation. We note that Hopper et al. (2004) did interpret the ultra-distal small blocks on Flemish Cap as oceanic crust but they also suggested that the root zone of the S detachment fault could be below the Flemish Cap margin. The kinematic at breakup as proposed in our model result is not incompatible with a root zone of the subcrustal Galicia S-detachment fault beneath Flemish Cap but not modelled in our results as explained in our reply to comment n°10 after the first review. We also note that our interpretation explains the crustal fault scarp at 0 km on Flemish Cap side and the enigmatic geometry East of the peridotite ridge at 0 km on Galicia Side (Fig. 2c and 6i).

We added the following two sentences line 244:

“We note that detachment 1 post-dates the formation of the sub-crustal S-detachment (Fig. 4b) that requires a root zone beneath Flemish Cap margin^{6,9,49}. The origin of the formation of the small continental blocks at the ultra-distal Flemish Cap margin proposed here (Fig. 6h,i) differs from previous interpretations^{6,49}.”

Two details: 1) maybe check again the consistency of magma-poor/a-magmatic/non-volcanic throughout the text;

Thank you about this comment. We have checked the consistency of usage of these terms and have modified the text accordingly. We note that we use “a-magmatic” term when writing about model results and “magma-poor” when referring to the Galicia-Flemish Cap rifted conjugate passive margins system.

2) Caption of Figure 2 refers to Figure S5 for full time evolution, it is Fig. S7.

We have modified the text accordingly.

Wishing to the authors good luck to finalize the MS.

Best regards,

Gaël Lymer

PS: My apologize for the delay in my review, I have got the Covid last week that prevented me to review this MS in time.

Thank you for the work done during the review process.

Reviewer #3 (Remarks to the Author):

As the authors point out, their manuscript is the first manuscript that really focus on how to make flip flop in a numerical model of spreading. This is of a wide interest outside the modelling community as it provides new constrain on sea floor spreading in magma poor conditions. In the first shorter version of the ms, they were not providing a complete parametric study which was weakening their point. This has been corrected in this new, longer version, both in the main body of the paper and in the supplementary material. They have adress all my point and I thank them for that. These addings strengthen sufficiently their conclusions for the paper to be published with no further revisions from my modeller point of view.

Laetitia Le Pourhiet

We appreciate the constructive comments of the reviewer that helped us to improve our manuscript.